# *Plasmodium vivax* epidemiology in Ethiopia 2000-2020: A systematic review and meta-analysis

**Tsige Ketema**[1,2]*, **Ketema Bacha**[1], **Kefelegn Getahun**[3], **Hernando A. del Portillo**[2,4,5], **Quique Bassat**[2,5]

**1** Jimma University, College of Natural Sciences, Department of Biology, Jimma, Ethiopia, **2** ISGlobal, Institute for Global Health, Hospital Clinic-Universitat de Barcelona, Barcelona, Spain, **3** Jimma University, College of Social Sciences and Humanity, Department of Geography and Environmental Studies, Jimma, Ethiopia, **4** IGTP, Germans Trias i Pujol Health Research Institute, Badalona, Spain, **5** ICREA, Catalan Institution for Research and Advanced Studies, Barcelona, Spain

* tsigeketema@gmail.com

## Abstract

### Background

Ethiopia is one of the scarce African countries where *Plasmodium vivax* and *P. falciparum* co-exist. There has been no attempt to derive a robust prevalence estimate of *P. vivax* in the country although a clear understanding of the epidemiology of this parasite is essential for informed decisions. This systematic review and meta-analysis, therefore, is aimed to synthesize the available evidences on the distribution of *P. vivax* infection by different locations/ regions, study years, eco-epidemiological zones, and study settings in Ethiopia.

### Methods

This study was conducted in accordance with Preferred Reposting Items for Systematic Reviews and Meta Analyses (PRISMA) guidelines. Studies conducted and published over the last two decades (2000 to 2020) that reported an estimate of *P. vivax* prevalence in Ethiopia were included. The Cochrane Q ($\chi^2$) and the I[2] tests were used to assess heterogeneity, and the funnel plot and Egger's test were used to examine publication bias. A p-value of the $\chi^2$ test <0.05 and an I[2] value >75% were considered presence of considerable heterogeneity. Random effect models were used to obtain pooled estimate of *P. vivax* infection prevalence. This study is registered with PROSPERO (International Prospective Register of Systematic Reviews): ID CRD42020201761.

### Results

We screened 4,932 records and included 79 studies that enrolled 1,676,659 confirmed malaria cases, from which 548,214 (32.69%) were *P. vivax* infections and 1,116,581 (66.59%) were due to *P. falciparum*. The rest were due to mixed infections. The pooled estimate of *P. vivax* prevalence rate was 8.93% (95% CI: 7.98–9.88%) with significant heterogeneity ($I^2$ = 100%, p<0.0001). Regional differences showed significant effects

**Data Availability Statement:** All relevant data are within the manuscript and its Supporting Information files.

**Funding:** The author (s) received no funding for this work.

**Competing interests:** The authors have declared that no competing interests exist.

(p<0.0001, and $I^2$ = 99.4%) on the pooled prevalence of *P. vivax*, while study years (before and after the scaling up of interventional activities) did not show significant differences (p = 0.9, $I^2$ = 0%). Eco-epidemiological zones considered in the analysis did show a significant statistical effect (p<0.001, $I^2$ = 78.5%) on the overall pooled estimate prevalence. Also, the study setting showed significant differences (p = 0.001, and $I^2$ = 90.3%) on the overall prevalence, where significant reduction of *P. vivax* prevalence (4.67%, 95%CI: 1.41–7.93%, p<0.0001) was observed in studies conducted at the community level. The studies included in the review demonstrated lack of publication bias qualitatively (symmetrical funnel plot) and quantitatively [Egger's test (coefficient) = -2.97, 95% CI: -15.06–9.13, p = 0.62].

## Conclusion

The estimated prevalence of *P. vivax* malaria in Ethiopia was 8.93% with *P. vivax* prevailing in the central west region of Ethiopia, but steadily extending to the western part of the country. Its distribution across the nation varies according to geographical location, study setting and study years.

## Author summary

*Plasmodium vivax* is the most widely distributed parasite worldwide. But it is a rare malaria parasite in Africa, except in the eastern part of the region. Ethiopia is one of the few countries in Africa where the two principal human malaria parasite, *P. falciparum* and *P. vivax* co-exist. Finding of the current review showed that a pooled estimate prevalence of *P. vivax* was 8.93% with significant heterogeneity. The prevalence was varied across different regions in the country, eco-epidemiological zones and study settings, where the highest prevalence was documented in the South Nations and Nationalities Peoples' Region, highlands at an altitude of 2000-2500masl and at health facilities, respectively, while study years (before and after the scaling up of malaria interventional activities) didn't show any effect on the pooled estimate prevalence of *P. vivax*. Overall, *P. vivax* showed high prevalence in the western central region of the country, but gradually spreading to the far-western part, previously assumed to be free of malaria. The spread of malaria in general and *P. vivax* in particular to malaria free regions could have far reaching consequences and calls for periodic surveillance of the disease to curb the potential public health risks.

## Introduction

*Plasmodium vivax* is one of the five human malaria parasites, with wider distribution across the globe [1]. It causes recurring malaria and affects a large number of populations globally [2]. Although it is widely accepted that the human *P. vivax* parasite has African origins [3], its presence in this continent has been unevenly distributed, and its clinical impacts are considered minor except in Eastern Africa [4]. Indeed, the horn of Africa (Ethiopia, Djibouti, Eritrea, and Somalia), South Sudan and the island of Madagascar seem to be the only countries where *P. vivax* is considered endemic and causes significant clinical disease in a stable manner, although reports from many other African countries confirm that the parasite does circulate

beyond this region. Such a disparate distribution of clinical disease is probably linked to the higher prevalence in these countries (and its generalized absence in the rest of the continent) of Duffy positive individuals, given that this species is thought to require the Duffy receptor to invade reticulocytes and cause disease [5]. However, for the past decade, the increasing demonstration of *P. vivax* associated infections and diseases in Duffy-negative individuals from a variety of West African countries [6, 7] confirm the underlying widespread presence of this species across other malaria-endemic regions of Africa, and the possibility that *P. vivax* has evolved to find an alternate ways of infecting the reticulocytes and causing disease [8]. Although this phenomenon is yet not widespread, it could further complicate achieving the current malaria elimination goals in the continent [7].

There are additional important knowledge gaps regarding *P. vivax*. The parasite's biology and its pathophysiology are still poorly understood, compared to that of *P. falciparum*. Current understanding of the hypnozoite and its basic biology remains elusive, and this is a critical gap that hampers current therapeutic and diagnostic strategies. Moreover, the early release of gametocytes to the bloodstream from the liver, even prior to the appearance of clinical symptoms, facilitates transmission, and obstructs control of this species. Such challenges significantly hamper current global *P. vivax* malarial control efforts, and calls for well-coordinated wider ranging research, surveillance and re-mapping of its global epidemiology [9].

Ethiopia accounts for 6% of the malaria cases globally, and about 12% of the global cases and deaths due to *P. vivax* [10]. The country has made significant efforts to control malaria since the introduction of dichlorodiphenyl-trichloroethane (DDT) as insecticide upon which the country based its indoor residual spraying (IRS) strategy back in 1959 [11, 12]. Several attempts have been made to scale up major malaria interventional activities such as the distribution of insecticide treated bed nets (ITN), indoor residual spraying (IRS), and introduction of artemisinin-based combination therapy (ACT) starting from 2005 [13]. As a result of these concerted efforts, in areas with Annual Parasite Incidence (API) of > 100 per 1,000 population (high transmission), significant reductions of API (from 14.3 per 1,000 in 2013 to 6.4 in 2016 per 1,000 population) were documented [14]. However, in low transmission areas, the API appeared to increase from 22.5 to 37.4 per 1000 population from 2013 to 2016 [14].

In Ethiopia, where the burden of *P. vivax* seems to be slowly rivalling that of *P. falciparum*, no attempt has been made to derive a robust epidemiological review of the *P. vivax* data available in the country. Clear understanding of the distribution of *P. vivax* is essential for informed decisions on appropriate control strategies to be designed and implemented against this neglected species. Thus, the main aim of this review was to synthesize evidence on distribution of *P. vivax* infection among symptomatic and asymptomatic cases in Ethiopia.

## Methods

### Research design

The study was conducted according to Preferred Reposting Items for Systematic Reviews and Meta Analyses (PRISMA) guidelines. The protocol was registered at PROSPERO International prospective register of systematic reviews, with ID: CRD42020201761 (available at: https://www.crd.york.ac.uk/PROSPERO/display_record.php?RecordID=201761).

### Search strategy

Potentially relevant articles were identified from PubMed (n = 1021), Embase (n = 1250), Web of Science (Core Collection) (n = 1356) and Scopus (n = 1298) electronic databases (Fig 1). A full search strategy for each database was developed using MeSH and free-text words to capture articles measuring *P. vivax* prevalence in Ethiopia in human without language restriction

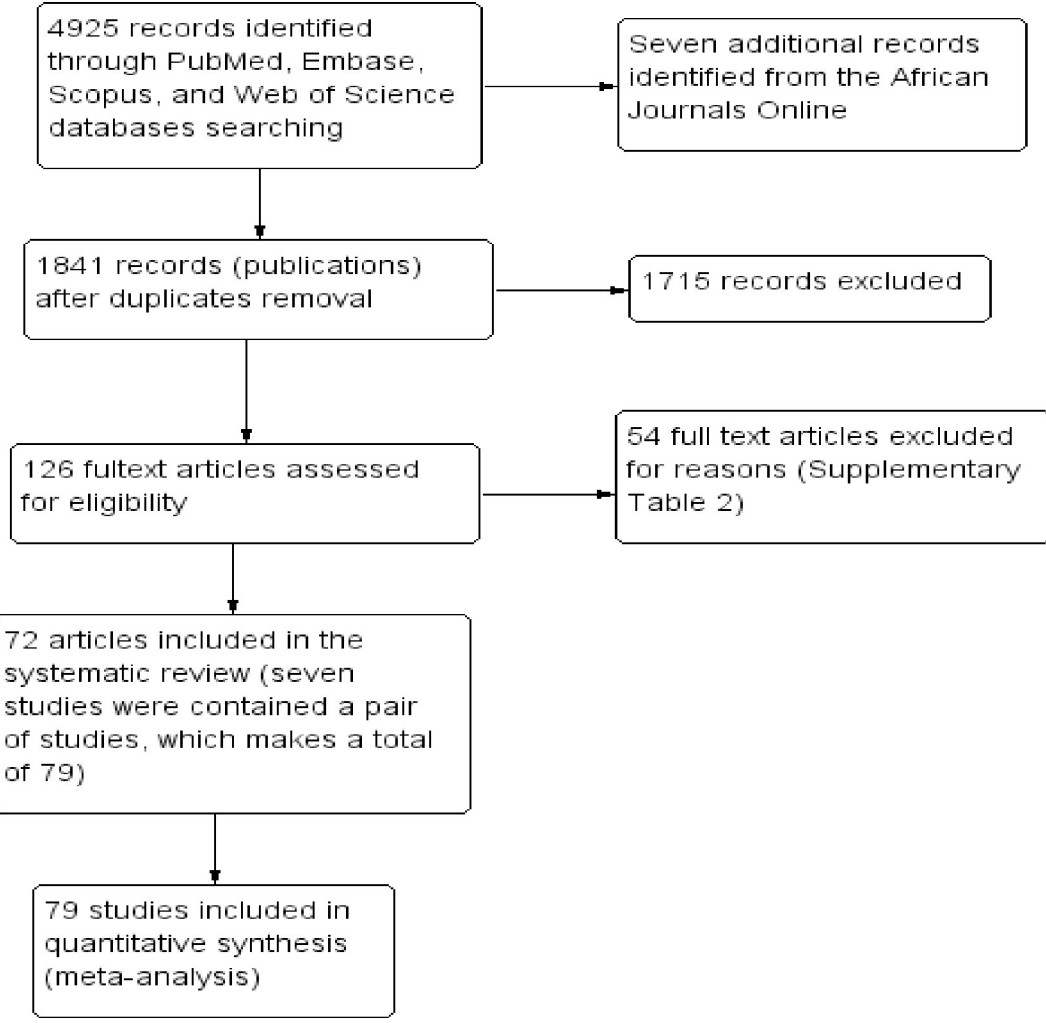

**Fig 1. Study flow diagram.**

(**see S1 Table** for the full detailed search strategies). Each search strategy was applied to articles published between 2000 and 2020. The last search was performed on 31[st] December 2020. In addition, an effort was made to retrieve more information manually from African Journal Online (AJOL) indexed journals (n = 7). Grey literature and non-published data were not included in the review. Results from different database searches were exported to EndNote and then combined followed by trimming out of any duplicated data.

## Eligibility criteria

Studies were eligible for inclusion if they were original publications describing the epidemiology of *P. vivax* in humans in Ethiopia. We included observational studies (cross-sectional and retrospective) written in any language and published over the last twenty years (from 1[st] January 2000 to December 31st 2020). Studies conducted both in health facilities (i.e., health posts, health centers, and hospitals) and at the community level (i.e., villages, and schools) were included. Other data sources such as reviews, conference abstracts, commentaries, editorials, registered protocols for clinical trials, letters to the editor, personal opinions, non-human or in

vitro studies, studies on other *Plasmodium* species and those with incomplete information (studies lacking data on prevalence of *P. vivax*) were excluded.

### Study selection

Two authors (TK and KB) independently screened titles and abstracts of all records identified by the search strategy for potential inclusion in the review. Afterwards, full-text copies of articles deemed potentially relevant were retrieved and their eligibility was assessed. Disagreements between individual judgments were resolved through discussion. We listed all studies excluded after full-text assessment and reasons for the exclusion (**S2 Table**).

### Data extraction

Two authors (TK and KB) used a data extraction form to independently extract data on study characteristics, including: type of study (facility or community based), age group, and presence or absence of symptoms. Additional information collected included study year (before or after the scale up of national malaria interventional activities) [14], geographical regions, diagnostic methods used, sample size, and the main characteristics of the population under study.

*Outcome of interest was p*revalence of *P. vivax* infection. *P. vivax* malaria diagnosis required parasitological confirmation irrespective of the methods used (optic microscopy, RDT, PCR, LAMP, ELISA, etc.). Original authors were contacted when further clarification and additional data were necessary.

### Assessment of risk of bias in included studies

The risk of bias for each included study was assessed independently by two authors (TK and KB) using the Prevalence Critical Appraisal Instrument, designed to be used in systematic reviews addressing questions of prevalence, as described by Munn et al. [15]. This tool assesses the methodological quality of studies reporting prevalence data using ten critical appraisal criteria: sample representation of the target population, participant recruitment appropriateness, sample size adequacy, subjects and setting detailed description, enough coverage of the identified sample, objectivity and standardization in the measurement of the condition, reliability in the measurement of the condition, statistical analysis appropriateness, confounders/ subgroups/differences identification and accounting, and subpopulations identification using objective criteria. An overall low ($\geq$7/10), medium (between 5 and 7/10), high (<5/10) risk of bias level was assigned to each study.

### Data synthesis and analysis

Data were analyzed using the Cochrane Review Manager (version 5.4) for qualitative and quantitative synthesis. Prevalence for each study was reported. For cases where prevalence was not reported, authors calculated it by dividing the event (*P. vivax* positive and/or in mixed infection) to the total population sampled in each study. Standard error of the mean (SE) for each study was calculated from the standard deviation obtained using the formula, $StDev = \sqrt{p(1-p)}$ where **p** is a proportion of the population with the event. Then, SE was calculated from the *StDev* using the formula, $SE = StDev\sqrt{n}$, where n is the sample size.

Heterogeneity between studies was evaluated using Cochrane's Q ($\chi^2$) and the $I^2$ tests. For the Cochrane's test, a p-value of the $\chi^2$ test less than 0.05 was considered as significant statistical heterogeneity. $I^2$ values of 25%, 50% and 75% were assumed to represent low, medium, and high heterogeneity, respectively. Outliers that might cause heterogeneity and meta-coefficient

were analyzed using Comprehensive Meta-analysis (CMA) software and presented using box plots (**S1 Fig**) and Table, respectively.

Subgroup analysis was conducted to investigate heterogeneity. Pre-specified subgroups potentially assumed to affect the overall prevalence estimate included: i) geographical location/regions (in Ethiopia there are currently ten regional states and two chartered cities), ii) study setting, iii) eco-epidemiological zones (altitude), and iv) study year. Likewise, due to high heterogeneity ($I^2 > 75\%$, $P < 0.05$), random effects models were used for the pooled statistics. Forest plots were used to display point estimates and confidence intervals. Publication bias for studies included in the meta-analysis was assessed quantitatively using the Egger's test and qualitatively constructing funnels plot and looking for asymmetry. ArcGIS software version 10.0 was used to sketch a map for the distribution of *P. vivax* malaria in the country.

## Results

### Study selection

A total of 4932 citations were initially identified. After the duplicates were excluded, 1841 unique citations were screened and assessed for eligibility. From the remaining 1841 screened at title/abstract level, 1715 records considered irrelevant for the purposes of the study were excluded. At the second phase of records assessment, a total of 126 eligible studies with available full text were thoroughly reviewed and a total of 72 articles (seven of them were comprised of a pair of an independent studies, which makes the total of studies 79) included for qualitative and quantitative meta-analysis, respectively (Fig 1). Detailed reasons for the 54 excluded studies are presented in **S1 Table**.

### Quality assessment of individual studies

Across the 10 quality domains evaluated, the majority of the studies met five or more of the quality criteria. Most of the studies (n = 31) met 8 or more of the quality criteria assessed, and others (n = 26) met 5 to 7 of the quality criteria assessed for prevalence studies. Only 15 studies were rated below 5 for the quality assessment. The most common quality criteria not fulfilled by the studies were: poor statistical analysis such as failure to use reliable, valid and appropriate data analysis tools (n = 27), failure to identify confounders/differences accounting (n = 24) and unclear sample recruitment (n = 19). Most of the studies fulfilled the following quality criteria: contained adequate sample size (n = 64), described the study subjects and setting in detail (n = 62), and the data analyses were conducted with sufficient coverage of the identified samples (n = 69). Nine studies met all 10 quality assessment criteria. Twenty-eight studies were based on data extracted from patients' medical records accessed from health facilities. For such studies, some of the quality criteria such as defining target population, use of appropriate sampling techniques and standard data collection tools/methods were difficult to evaluate and were considered as not applicable (NA) (**S3 Table**).

### Study characteristics

A total of 72 articles, but 79 studies, were finally included in the meta-data analysis, 18 studies have reported data from 8 study sites (more than one study from single site), at different years and seasons, and by different authors using different study populations. They reported on prevalence data from the following towns: **Arbaminch** [16–18], **Arijo Didhesa** [19, 20], **West Armachew** [21, 22], **Butajira** [23, 24], **Dore Bafeno** [25, 26], **Jimma town** [27, 28], **Wolkite**

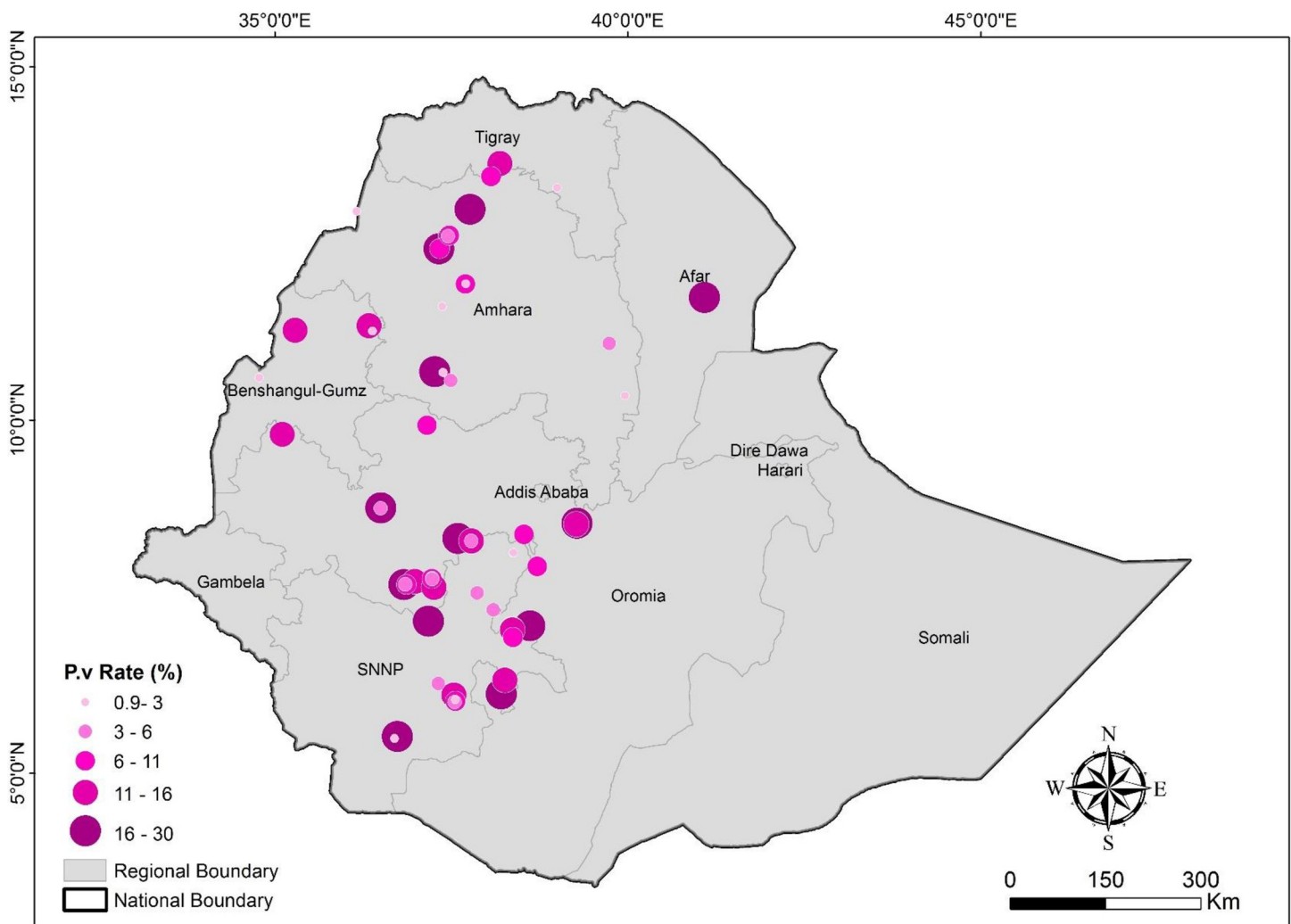

**Fig 2. Map showing estimates of *P. vivax* prevalence from the 72 study sites according to geographical distribution in Ethiopia.** The size of the purple dots is proportional to the prevalence estimates reported. The map was sketched by one of the authors using ArcGIS software.

[29, 30], and **Woreta** [31, 32]. The rest of the studies typically reported data from a single study site, although some reported data for multiple seasons (Fig 2).

Twenty-eight studies reported pooled prevalence data based on retrospective evaluations of 5–20 years' patient data collected from health facilities. The remaining 51 were cross sectional studies undertaken at health facilities (n = 60) or at the community level (n = 19). Malaria diagnosis relied on optic microscopy in the majority of studies (n = 60/79, 75.95%); with the remaining 19 studies using either only RDT (n = 3), microscopy plus RDT (n = 11), microscopy plus PCR (n = 2), a mix of the three techniques (microscopy, RDT and PCR; n = 3). Participants of most of the included studies (n = 59/79, 74.7%) were all-age groups populations, while 11 were from children and teenagers up to 15 years of age, five studies included population aged >15 years and four studies enrolled only pregnant women. The 79 studies enrolled a total of 5,930,976 study participants (ranging from 178 to 2,827,722) among which 1,676, 659 were malaria positive. A total of 548,214 participants [about 9.24%, (ranging from 1 to 267,242)] had a confirmed *P. vivax* infection [mono infection (n = 525,674; 95.9%) and mixed infection (n = 22,406; 4.1%)] [16–86]. Ethiopia is a federal state https://en.wikipedia.org/wiki/

Federation subdivided into ethno-linguistically based regional states. There are currently ten regional states and two chartered cities. In line with this division, the studies reported data from the regions of Afar (n = 1), Amhara (n = 26), Benishangul (n = 3), Oromia (n = 18), Southern Nations, Nationalities and Peoples' Region (SNNPR) (n = 25), Tigray (n = 1), Harari (n = 1) and nationwide surveys of Ethiopia (n = 4). Accordingly, the majority of the malaria research reports (69/79, 87.34%) presented data from Amhara, Oromia and SNNPR. Based on the eco-epidemiological zones of malaria distribution, 22 studies were reported from areas with altitude <1500m (low lands with seasonal/intense transmission), 10 were from altitudes between 1500-1750m (high land fringe, high unstable transmission), 14 were from altitudes ranging between 1750-2000m (high land fringe, low unstable transmission), 7 studies were from districts with altitudes of 2000-2500m (highland, occasional epidemic) and 23 were from areas with mixed ecological zones (Table 1), and three studies without this information were excluded [48, 49].

## Main outcome of the meta-analysis

The overall random effects pooled prevalence rate of *P. vivax* (mono-infection and mixed infection with *P. falciparum*) in Ethiopia was 8.93% (95% CI: 7.98–9.88%), with a very high level of heterogeneity ($I^2$ = 100%, p<0.0001). Indeed, the prevalence of *P. vivax* across individual studies varied considerably [ranging from 0.25, n = 1/400 among all age groups in SNNPR [85] to 47.35%, n = 197/416 in all age groups in many sites throughout Ethiopia using 18r based nested PCR [74] (Fig 3).

The pooled prevalence of *P. vivax* in mono-infection was 7.98% (95% CI: 7.09–8.87%) with a very high level of heterogeneity (Fig 4) and prevalence of *P. vivax* in a mixed infection (*P. vivax* with *P. falciparum)* was 0.73% (95% CI: 0.65–0.82%). The prevalence reported in each study for mixed infection was also varied and ranged from 0.005% [51] to 7.9% [74] (Fig 5). Analysis of risk of publication bias among the studies included in the current review showed there was no publication bias as demonstrated by asymmetrical funnel plot qualitatively (S2 Fig) and non-significant Egger's regression test quantitatively (bias coefficient = -2.97, 95% CI: -15.06 to 9.13, p = 0.62). Two of the studies included had far-out values (47%) and outside values (30%) [Coefficient of Skewness = 1.81, p<0.001] (**S1 Fig**).

Regional variation showed significant effect on the estimated prevalence of *P. vivax* although there was high significant heterogeneity ($I^2$ = 100%, p<0.0001) within each of the three main regions (Amhara, Oromia and SNNPR). SNNPR is a region where significantly highest (10%, 95%CI: 8.46–11.54%) pooled prevalence of *P. vivax* is documented (**S3 Fig**). Three studies (one of them contained a pair of studies) included in the review, which reported national/regional or more than one region prevalence were excluded from the locations/region's analysis [58, 86, 87] (**S3 Fig**).

The different eco-epidemiological zones considered in the meta-analysis did appear to significantly affect the pooled estimate prevalence of *P. vivax* ($\chi^2$ = 18.65, df = 4, p = 0.0.01, $I^2$ = 78.5%). Moreover, some studies reported from the highlands with occasional malaria epidemic zones (2000-2500m) contributed to the observed high prevalence of *P. vivax* (9.80%, 95%CI: 6.73–12.87%) compared to other eco-epidemiological zones (**S4 Fig**).

There were significant study setting differences (facility and community) among the studies ($\chi^2$ = 10.27, df = 1, p = 0.001, and $I^2$ = 90.3%). Being diagnosed and treated at the health facility (health centers, health posts and hospitals) significantly (10.44%, 95%CI: 9.09–11.79%, p<0.0001) affected the overall pooled prevalence of *P. vivax*, although there was substantial unexplained high heterogeneity within the studies conducted at both settings ($I^2$ = 100% for both). Hence, the validity of study setting effect estimate for each subgroup is uncertain as

**Table 1. Characteristics of the studies included in the epidemiological studies of *P. vivax* in Ethiopia (2000–2020).**

| Author ID | Study site/City/district | Region | Altitude (m) | Setting | Study design | Study year/period | Sample tested | Study population Key characteristics | Age | Gender | Diagnostic method | Malaria positive | P. falciparum | P. vivax | Mixed infection | Group |
|---|---|---|---|---|---|---|---|---|---|---|---|---|---|---|---|---|
| Abossie et al., 2020 | Arbaminch | SNNPR | 1,285 | Health facility | Cross-sectional | April 2017—May 2017 | 271 | Febrile children. Exclusion if antimalarial drug administration up to 3 months prior to the study | Range: 12–59 months; Mean: 31.2 months | 58% males, 42% females | Microscopy | 60 | 30 | 29 | 1 | Children |
| Addisu et al., 2020 | Gorgora and Chuahit in Dembia district. | Amhara | 1,850–2,000 | 2 health facilities | Retrospective clinical record review | 2012–2018 | 11,879 | Patients that were requested a blood film | All ages | 57% males, 43% females | Microscopy | 2590 | 1756 | 733 | 101 | All ages |
| Alelign et al., 2018 | Woreta town, Fogera district | Amhara | 1828 | Health facility | Retrospective clinical record review | 2005–2012 | 102,520 | Suspected cases of malaria | All ages | 53% males, 47% females | Microscopy | 33431 | 23274 | 8870 | 1287 | All ages |
| Alemayehu et al., 2015 | Diverse | Oromia | Mix | 12 health facilities | Cross-sectional | Sept 2011—Nov 2011 | 1,819 | HIV-positive patients having routine follow-up visits at HIV care and treatment clinics | ≥ 18 years | 36% males, 64% females | Microscopy | 13 | 6 | 7 | ND | ≥ 18 years |
|  |  |  |  |  |  |  | 1,819 | HIV-sero-negative patients attending the general medical outpatients departments | ≥ 18 years | 54% males, 46% females | Microscopy | 143 | 69 | 74 | ND | ≥ 18 years |
| Alemu & Mama, 2018 | Arbaminch | SNNPR | 1,285 | Blood bank | Cross-sectional | Feb 2015—June 2015 | 416 | Blood donors, asymptomatic. Exclusion of permanent residents of known non-endemic malaria areas | Range:18–59 years; Median: 22 years | 56% males, 44% females | Microscopy | 17 | 8 | 9 | ND | ≥ 18 years |
| Alemu et al., 2011 | Jimma town | Oromia | 1,750 | Community, house-hold-based survey | Cross-sectional | April 2010—May 2010 | 804 | Households' residents | All ages; Median: 21 (SD 1.2) years | 42% males, 58% females | Microscopy | 42 | 11 | 30 | 1 | All ages |
| Alemu et al., 2012b | Azezo | Amhara | 1,400 | Health facility | Cross-sectional | Feb 2011—March 2011 | 384 | Febrile patients. Exclusion of pregnant women, if known concomitant chronic infections, or if antimalarial drug administration in the 2 weeks prior to the study | Range: 1–80 years; Median: 23.8 years | 51% males, 49% females | Microscopy | 44 | 9 | 33 | 2 | All ages |
| Alemu et al., 2014 | Dabat district | Amhara | Mix | 4 health facilities | Cross-sectional | August 2012—May 2013 | 1,644 | Residents visiting local health centers | All ages | ND | Microscopy or RDT | 645 | 355 | 173 | 117 | All ages |
| Alkadir et al., 2020 | Mankush | Benshangul | ND | Health facility | Retrospective clinical records review | Jan 2014—Dec 2018 | 16,964 | Malaria suspects | All ages | ND | Microscopy | 8658 | 6513 | 2121 | 24 | All ages |
| Animut et al., 2009 | Dembecha, Jiga, Gebeze Mariam, Finoteselam | Amhara | ND | 4 health facilities | Cross-sectional | Sep 2006—Nov 2006 | 653 | Febrile outpatients. Exclusion of children requiring inpatient treatment or with chronic disease | Range: 3–17 years; Median: 8.4 years | 51% males, 49% females | Microscopy | 506 | 309 | 150 | 47 | All ages |
| Argaw et al. 2016 | Diverse | Mix | Mix | 110 health facilities | Retrospective clinical records review | April 2012—Sep 2015 | 873,707 | Malaria suspected patients with a diagnostic test result | All ages | 60% males, 40% females | Microscopy and RDT | 223,293 | 108704 | 96765 | 8790 | All ages |

*(Continued)*

**Table 1.** (Continued)

| Author ID | Study site/ City/district | Region | Altitude (m) | Setting | Study design | Study year/ period | Sample tested | Study population — Key characteristics | Age | Gender | Diagnostic method | Malaria positive | P. falciparum | P. vivax | Mixed infection | Group |
|---|---|---|---|---|---|---|---|---|---|---|---|---|---|---|---|---|
| Aschale et al., 2018 | West Armachiho district | Amhara | 667 | Community, 10 farm sites | Cross-sectional | Sep 2016—Dec 2016 | 385 | Asymptomatic migrant laborers | Range: 15–60 years; Mean: 26.3 (SD 8.9) years | 90% males, 10% females | Microscopy | 71 | 50 | 7 | 14 | ≥15 years |
| Aschale et al., 2019 | West Armachiho district | Amhara | 667 | Community, 11 farm sites | Cross-sectional | Oct 2016—Dec 2016 | 178 | Migrant laborers. Exclusion if taken medication for malaria and/or visceral leishmaniasis for the last 2 weeks | Range: 15–65 years; Mean 26.1 (SD 8.6) years | 92% males, 8% females | Microscopy | 40 | 29 | 4 | 7 | ≥15 years |
| Ashton et al. 2011 | Diverse | Oromia | Mix | Community, school-based survey (197 schools) | Cross-sectional | May 2009, Oct 2009-Dec 2009 | 20,899 | Children. Excluded if the blood film was missing or unreadable | Range: 5–18 years; Median 11 (IQR: 9–12). | 53% males, 47% females | Microscopy[1] | 117 | 61 | 55 | 1 | Children |
| Assefa et al., 2015 | Hossana | SNNPR | 2,177 | Health facility | Cross-sectional prior to an RCT | April 2014 | 1,693 | Clinically malaria-suspected individuals with fever or history of fever seeking treatment | All ages | ND | Microscopy | 281 | 182 | 92 | 7 | All ages |
| Awoke & Arota, 2019 | Tercha Hospital | SNNPR | 1406 | Facility | Cross-sectional | March 20 to May 30, 2016. | 340 | All acute febrile patients clinically suspected of malaria | Range: 15–50 years; Mean 27.6 | 68% males, 32% females | Microscopy | 170 | 105 | 61 | 4 | All ages |
| Ayalew et al., 2016 | Jiga area | Amhara | 1,812 | Community, household-based survey | Cross-sectional | Nov 2013—Dec 2013 | 392 | Households' residents (one person randomly selected per household) | Range: 1–80 years; Mean 21.9 | 38% males, 62% females: 9% self-reported pregnant | RDT | 11[2] | 6 | 5 | 0 | All age |
| Belete and Roro, 2016 | Chichu, Wonago | SNNPR | 1,650 | Health facility | Cross-sectional | May 2016—June 2016 | 324 | Outpatients with history of fever in the last 24h. Exclusion if not resident or anti-malarial treatment during the previous 8 days | All ages | 53% males, 47% females | Microscopy | 91 | 32 | 48 | 11 | All ages |
| Birhanie et al., 2014 | Dembia district | Amhara | 1,750–2,100 | Health facility | Cross-sectional | April 2013—May 2013 | 200 | Febrile patients suspected for malaria and/or typhoid fever. Exclusion if antimalarial treatment and/or antibiotics within the previous 2 weeks | Range: 2–80 years; Mean 24.2 (SD: 13.4) | 60% males, 40% females | Microscopy | 73 | 32 | 30 | 11 | All age |
| Beyene et al., 2020 | Jardga Jarete district | Oromia | 1,400–2,700 | 3 health facilities | Retrospective clinical records review | 2015–2019 | 25,868 | Malaria suspects. Excluded if malaria diagnosis results were not properly documented | ≥ 1 year | 60% males, 40% females | Microscopy | 4,336 | 2,561 | 1434 | 342 | All age |
| Dabaro et al., 2020 | Boricha district | SNNPR | 1001–2076 | 51 Health facilities | Retrospective clinical records review | 2010–2017 | 135,607 | Malaria suspects. Exclusion if incomplete record | All ages | 51.4% males, 48.4% females | Microscopy or RDT | 29,554 | 16,647 | 11,360 | 1,547 | All ages |
| Debo & Kassa, 2016 | Benna Tsemay district | SNNPR | 1,500 | Community, household-based survey | Cross-sectional | Dec 2011—Jan 2012 | 461 | Household residents of pastoralist communities | Range: 9 months– 65 years; Median: 13 years | 48% men, 52% female (7% pregnant,7.5% lac tating) | Microscopy or RDT | 28 | 18 | 6 | 4 | All ages |

(*Continued*)

**Table 1.** (Continued)

| Author ID | Study site/City/district | Region | Altitude (m) | Setting | Study design | Study year/period | Sample tested | Study population Key characteristics | Age | Gender | Diagnostic method | Malaria positive | P. falciparum | P. vivax | Mixed infection | Group |
|---|---|---|---|---|---|---|---|---|---|---|---|---|---|---|---|---|
| Degarege et al., 2011 | Dore Bafeno | SNNPR | 1,708 | Health facility | Cross-sectional | January, 2010 | 269 | Malaria suspects. Exclusion if anti-malarial treatment within the previous 2 weeks | All ages | 53.5% males, 46.47% females | Microscopy | 178 | 146 | 28 | 4 | All ages |
| Degarege et al., 2012 | Dore Bafeno | SNNPR | 1,708 | Health facility | Cross-sectional | Dec 2010—Feb 2011 | 1,065 | Malaria suspects. Exclusion if anti-malarial treatment within the previous 2 weeks | Range: 1-82 years; Mean 18.6 years | 51% males, 49% females | Microscopy | 306 | 138 | 154 | 14 | All ages |
| Delil et al, 2016 | Hadiya zone | SNNPR | 2,106 | 12 health facilities | Cross-sectional | May-June, 2014. | 411 | Febrile patients | Range: 18 years to 70 years, Mean 30.7 years | 50.4% males, 49.6% females | Microscopy | 106 | 27 | 76 | 3 | Adult >18 |
| Demissie and Ketema, 2016 | Mendi | Oromia | 1,538 | 2 health facilities | Cross-sectional | Sep 2014—June 2015 | 4,813 | Malaria suspects | Range: one month-60years, median age 14 years | ND | Microscopy | 1,434 | 851 | 533 | 50 | All ages |
| Derbie and Alemu, 2017 | Woreta | Amhara | 1,828 | Health facility | Retrospective clinical records review | Sep 2011—August 2012 | 8,057 | Malaria suspects. Exclusion if incomplete record | Range: 1-85 years; Median 25 years | 45% males, 55% females | Microscopy | 435 | 233 | 184 | 17 | All ages |
| Dufera et al., 2020 | Arjo Didhessa sugar cane plantation area | Oromia | 1275-1570 | Community, household-based survey | Cross-sectional | May 2016—Nov 2017 | 443 | Household's residents | All ages | ND | Microscopy | 14 | 6 | 8 | ND | All ages |
| | | | | Health facility | Retrospective clinical records review | 2013–2017 | 65,275 | Outpatients | All ages | ND | Microscopy | 4,164 | 776 | 3,170 | 218 | All ages |
| Ergete et al., 2018 | Salamago and Benatsemay districts | SNNPR | Mix | 2 health facilities | Retrospective clinical records review | Jan 2008—Dec 2014 | 54,160 | Malaria suspects with a blood smear | All ages | 61% males, 39% females | Microscopy | 22,494 | 13,727 | 7,297 | 1,470 | All ages |
| Esayas et al., 2020a | Kolla-Shara village | SNNPR | 1,170-1,390 | Community, household-based survey | Prospective (repeated cross-sectionals) | July 2016—Dec 2016 | 131 | Febrile household's residents. Individuals were screened twice per month for fever episodes | All ages | ND | RDT and microscopy confirmation | 46 | 27 | 19 | ND | All ages |
| Esayas et al., 2020b | Harari | Harari | 1552-1957 | Health facility | Retrospective clinical records review | 2013-2019 | 95,629 | Malaria suspected cases | All ages | ND | Microscopy or RDT | 44,882 | 28,576 | 12576 | 77 | All ages |
| Feleke et al., 2018 | Ataye | Amhara | 1,468 | Health facility | Retrospective clinical records review | 2013-2017 | 31,810 | Malaria suspects. Exclusion if record incomplete | All ages | ND | Microscopy | 2,670 | 2,087 | 557 | 26 | All ages |
| Feleke et al., 2020 | North-Shoa zone | Amhara | 1,532-1,788 | 3 health facilities | Cross-sectional | Nov 2018—Jan 2019 | 263 | Asymptomatic pregnant women. Exclusion if disease symptom/signs within the last 48h, treated with anti-malarial drugs in the previous 6 weeks, long-term medical treatment uptake or non-permanent resident in the area | Range: 16-41 years; Mean 27.8 (SD: 5.3) years | - | Microscopy[3] | 15 | 9 | 6 | 0 | Pregnant |
| Ferede et al., 2013 | Metema | Amhara | 685 | Health facility | Retrospective clinical records review | Sep 2006—Aug 2012 | 55,833 | Malaria suspects | All ages | 54% males, 46% females | Microscopy | 9,486 | 8,602 | 852 | 32 | All ages |

*(Continued)*

**Table 1.** (Continued)

| Author ID | Study site/City/district | Region | Altitude (m) | Setting | Study design | Study year/period | Sample tested | Study population Key characteristics | Age | Gender | Diagnostic method | Malaria positive | P. falciparum | P. vivax | Mixed infection | Group |
|---|---|---|---|---|---|---|---|---|---|---|---|---|---|---|---|---|
| Gebretsadik et al., 2018 | Kombolcha | Amhara | 1,875 | Health facility | Retrospective clinical records review | 2009–2016 | 27,492 | Malaria suspects. Exclusion of incomplete records | All ages | ND | Microscopy | 2,066 | 1,243 | 734 | 89 | All ages |
| Geleta and Ketema | Pawe district | Benishangul | 1050 | Health facility | Cross-sectional | October 2013 to May-2014 | 1523 | Malaria suspected cases | All ages | ND | Microscopy | 623 | 420 | 140 | 63 | All ages |
| Golassa & White, 2017 | Adama malaria diagnostic centre | Oromia | 1,712 | Health facility | Cross-sectional | May 2015–April 2016 | 3,161 | Malaria suspects | All ages | 68% males, 32% females | Microscopy | 1,141 | 326 | 847 | 32 | All ages |
| Gontie et al., 2020 | Sherkole district | Benishangul | 680–800 | Community | Cross-sectional | July 2018–August 2018 | 498 | Pregnant women. Exclusion if mental illness or severely debilitating disease | ≥ 15 year | - | RDT | 51 | 46 | 5 | ND | Pregnant women |
| Haile et al., 2020 | Dembecha | Amhara | 2,083 | Health facility | Retrospective clinical records review | Sep 2011–August 2016 | 12,766 | Malaria suspects. Exclusion of incomplete records. | All ages | 57% males, 43% female | Microscopy | 2,086 | 1,433 | 549 | 104 | All ages |
| Haji et al., 2016 | East Shewa zone | Oromia | 1,549–2,093 | 5 health facilities | Cross-sectional | Oct 2012-Nov 2012 | 830 | Malaria suspects | < 16 years; Mean: 6 years; Median: 6.1 years | 49% males, 51% females | Microscopy[4] | 170 | 70 | 97 | 3 | Children |
| Hassen & Dinka, 2020 | Batu town | Oromia | 1657 | Health facility | Retrospective clinical records review | 2012–2017 | 175423 | Malaria suspected cases | All ages | 53% males, 47% females | Microscopy | 21797 | 10791 | 11006 | ND | All ages |
| Hawaria et al., 2018 | Arjo-Didessa sugar development site | Oromia | 1300–2280 | Health facility | Retrospective review clinical records registers of 11 health facilities | 2008–2017 | 54020 | Malaria suspected cases | All ages | 64.5% males, 35.5% female | Microscopy, RDT | 18049 | 8660 | 7649 | 1740 | All ages |
| Ifa, 2018 | Konga Health Center | SNNPR | 2044 | Health facility | Retrospective clinical records review | 2011–2015 | 5210 | Malaria suspected cases | Children under five years | 51% males, 49% females | Microscopy | 2459 | 1402 | 1057 | ND | Children |
| Jemal and Ketema, 2019 | Asendabo town | Oromia | 1791 | Health facility | Retrospective clinical records review | 2007–2016 | 68421 | Malaria suspected cases | All ages | 52.5% Males, 47.5% females | Microscopy | 13624 | 7087 | 6508 | 29 | All ages |
| Kalil et al., 2020 | Bale zone | Oromia | Mix | Health facility | Retrospective clinical records review | January 2010-December 2017 | 62,392 | malaria suspected individuals who had visited the health facilities in Bale zone | All ages | 63% males, 37% females | Microscopy or RDT | 10,986 | 9,850 | 2036 | ND | All ages |
| Karunamoorthi & Bekele, 2009 | Serbo health center, Jimma zone | Oromia | 1740–2660 | Health facility | Cross-sectional | July 2007 and June 2008 | 6863 | Febrile patients presenting malaria symptoms | All ages | 64% males, 36% female | Microscopy | 3009 | 1946 | 1052 | 11 | All ages |
| Lankir et al., 2020 | Central, North and West Gondar zones | Amhara | Mix | Health facility | Retrospective clinical records review | July 2013–June 2018 | 2,827,722 | Malaria suspected cases | All ages | ND | Microscopy or RDT | 1,003,391 | 736,149 | 266,797 | 445 | All ages |
| Legesse et al., 2015 | Wolita zone | SNNPR | 2950 | Health facility | Retrospective clinical records review | 2008–2012 | 317,867 | Malaria suspected cases | All ages | 51% males, 49% female | Microscopy | 105,755 | 75,927 | 25,329 | 4497 | >15 years |
| Lo et al. 2015 | Six different localities across Ethiopia (Bure, Halaba, Asendabo, Jimma, Menkusha, Metehara, Shewarobit | Ethiopia | Mix | Community | Cross-sectional | ND | 390 | Asymptomatic individuals representing the younger age < 18 years and older age >18 years | All ages | ND | Nested PCR of the 18S rRNA region | 73 | 49 | 23 | 1 | All ages |
| | | | | Health facility | Cross-sectional | ND | 416 | Symptomatic or febrile patients visiting the health centres or hospitals | All ages | ND | Nested PCR of the 18S rRNA region | 331 | 134 | 164 | 33 | All ages |

(*Continued*)

**Table 1.** (Continued)

| Author ID | Study site/City/district | Region | Altitude (m) | Setting | Study design | Study year/period | Sample tested | Study population — Key characteristics | Age | Gender | Diagnostic method | Malaria positive | P. falciparum | P. vivax | Mixed infection | Group |
|---|---|---|---|---|---|---|---|---|---|---|---|---|---|---|---|---|
| Mekonnen et al., 2014 | Omo Nada, Bala Wajo and Arba Minch | Oromia, SNNPR | MiX | Health facility | Cross-sectional | August and December 2011 | 1416 | Self-presenting febrile patients attending health centres | All ages | 60.2% males, 39.8% females | Microscopy and PCR | 307 | 125 | 154 | 24[5] | All ages |
| Minwuyelet et al., 2020 | Gondar Zuria district | Amhara | 1750–2600 | Community | Cross-sectional | May- June 2019 | 251 | Individuals with clinical symptom of malaria and those taking antimalarial drugs 1 month prior to data collection excluded | All ages, mean: 24.6 years | 47% males, 53% females | Microscopy | 30 | 5 | 25 | ND | All ages |
| Nega et al. 2015 | Arbaminch town | SNNPR | 1,200–1,300 | Community | Cross-sectional | April and June, 2013 | 341 | Pregnant women without disease symptom/sign within the past 48 hours | ranged from 17 to 40 years with a median age of 25 | | Microscopy, or RDT | 31 | 12 | 15 | 4 | Pregnant women |
| Schicker et al., 2015 | Metema and west armachiho | Amhara | 717 | Community | Cross-sectional | 17–26 July, 2013 | 592 | a venue-based survey of 605 migrant laborers 18 years or older | >18 years, mean: 22.8 years | 98% males, 2% females | RDT | 71 | 57 | 10 | 4 | >18 years and above |
| Shambo and Petros, 2019 | Halaba special district | SNNPR | 1554 to 2149 | Health facility | Retrospective clinical records review | September 2013–August 2017 | 583668 | Malaria suspect cases | All ages | 49.8% males, 50.2% females | Microscopy | 55252 | 21397 | 33855 | ND | All ages |
| Shiferaw et al. 2018 | Tselemti District | Amhara | 1400 | Health facility | Retrospective clinical records review | January 2013 and December 2015 | 41773 | Malaria suspect cases | All ages | 54% males, 46% females | Microscopy | 11745 | 6835 | 4165 | 745 | All age |
| Solomon et al., 2020a | Wolkite health center Gurage zone | SNNPR | 1910–1935 | Health facility | Retrospective clinical records review | January 2015—December 2018 | 121230 | Malaria suspected cases | All ages, majority(54%) were >15 years | 51% males, 48.3% females | Microscopy | 10379 | 3044 | 7239 | 98 | All ages |
| Solomon et al., 2020(b) | Wolkite health center Gurage zone | SNNPR | 1910–1935 | Health facility | Cross-sectional | June 2019—August 2019 | 230 | asymptomatic pregnant women | >18 years, majority (72.2%) were between 18–27 years | - | Microscopy | 50 | 20 | 30 | ND | Pregnant women |
| Tadesse and Tadesse, 2013 | Felegeselam Health Center | Amhara | 1000–1050 | Health facility | Cross-sectional | December, 2011 | 398 | Acute febrile patients | All ages | 51% males, 48.2 Females, | Microscopy | 201 | 194 | 7 | ND | All ages |
| Tadesse et al., 2015 | Malo (Salayish Mender 4 and Tatta-qirchiqircho) | SNNPR | 591 | Community | Cross-sectional | February 2014, in the dry season | 555 | Asymptomatic Community members residing in the study sites for at least 2 years | All ages | | Microscopy, RDT, nested PCR | 54 | 29 | 24 | 1 | All ages |
| Tadesse et al., 2017 | Andassa, Yinessa, Ahuri, Yeboden, Fendika schools | Amhara | 1218–2010 | Community; five elementary schools | Cross-sectional | First survey June, 2015 | 555 | Students attending the elementary schools | Children, median age is 12 years | 51.3% males, 48.7% females | Microscopy, RDT, 18S based nested PCR, ELISA | 56 | 43 | 13 | ND | Children |
| | | | | | | second survey November 2015 | 294 | Students attending the elementary schools | Children, median age is 12 years | 51.3% males, 48.7% females | Microscopy, RDT, 18S based nested PCR, ELISA | 52 | 38 | 14 | ND | Children |
| Tesfa et al., 2018 | Adi Arkay health centre | Amhara | 1750–2100 | Health facility | Retrospective clinical records review | 1997–2013 | 20,483 | Malaria suspected cases | All ages | ND | Microscopy | 7392 | 5089 | 2128 | 173 | All age |

(*Continued*)

**Table 1.** (Continued)

| Author ID | Study site/City/district | Region | Altitude (m) | Setting | Study design | Study year/period | Sample tested | Study population Key characteristics | Age | Gender | Diagnostic method | Malaria positive | P. falciparum | P. vivax | Mixed infection | Group |
|---|---|---|---|---|---|---|---|---|---|---|---|---|---|---|---|---|
| Tesfaye et al., 2011 | Butajira district | SNNPR | 1900 | Community | Cross-sectional | October, November, and December, 2006 | 1082 | Members of two farming associations | >15 years old | 52% males, 48 females | Microscopy | 48 | 16 | 32 | ND | All ages |
| Tesfaye et al., 2019 | Tanquea Abergelle | Tigray | 1542 | Community | Cross-sectional | September 8 to October 18, 2017 | 1300 | Malaria suspected cases | All ages | 46.6% males, 53.4% females | Microscopy | 876 | 856 | 20 | 2 | All ages |
| Tuasha et al., 2019 | Kella, Aruma and Busa Health Centers in Wondo Genet | SNNPR | 1880 | Health facility | Cross-sectional | December 2009 to July 2010 | 427 | malaria suspected febrile patients from three health centers | ranged from 6 -77years (mean ± SD = 20.8 years | 55% males, 45% females | Microscopy | 276 | 202 | 71 | 3 | All ages |
| Woday et al., 2019 | Dubit district | Afar | 800–1000 | Health facility | Cross-sectional | April 15th to 15th May 2018 | 484 | All under-five children who presented with fever symptoms | Children, mean age was 28 months | 56.6% males, 43.4% females | Microscopy or RDT | 310 | 206 | 72 | 32 | children |
| Wondimeneh et al., 2018 | Kolla-Diba health center | Amhara | 2040 | Health facility | Cross-sectional | November 01, 2015 to May 30, 2015 | 384 | HIV positive febrile patients | All ages, mean age of 28 years | 59% males, 41% females | Microscopy | 53 | 8 | 4 | 0 | All ages |
| | | | | | | | | HIV negative febrile patients | All ages, mean age of 28 years | 59% males, 41% females | Microscopy | 79 | 43 | 31 | 5 | All ages |
| Woyessa et al., 2012 | Butajira area (six kebeles) | SNNPR | 1800–2300 | Community | Cross-sectional | October 2008 to June 2010 | 19,207 | all family members who consented to the study | Ranged: 0 months-99years, mean age was 20.5 years | 48.7% males, 51.3% females | Microscopy | 178 | 22 | 154 | 2 | All age |
| Yehualaw et al., 2009 | Gilgel-Gibe hydroelectric dam | Oromia | 1,734–1,864 | Community | Cross-sectional | October and December 2005 | 1855 | At risk Children those living in villages within 3 km of the reservoir | children under 10 years | 48.8% males, 51.2% females | Microscopy | 142 | 59 | 83 | ND | Children |
| | | | | | | | 774 | Control, Children living in villages within 5-8km from its shore | children < 10 years, mean age4.7 years | 48.7% males, 51.3% females | Microscopy | 51 | 17 | 34 | ND | Children |
| Yimer et al., 2015 | Walga, Borer, Jeju, and Nacha Qulit kebeles | SNNPR | 1100–2300 | Community | Cross-sectional | December 2013 | 400 | afebrile individuals residing in the visited house holds | All ages | 42% males, 58% females | Microscopy | 1 | 0 | 0 | 1 | All ages |
| | Walga Health Center Abeshge District, | SNNPR | 1100–2300 | Health facility | Retrospective clinical records review | February 2008 and December 2012 | 34,060 | Malaria suspected cases | All ages | 52% males, 48% females | Microscopy | 11523 | 5889 | 5489 | 150 | All ages |
| Yimer et al., 2017 | Felegehiwot referral Hospital | Amhara | 1840 | Health facility | Retrospective clinical records review | 2010–2014 | 14,750 | Malaria suspected cases | All ages | 50.3% males, 49.7% females | Microscopy | 740 | 397 | 331 | 12 | All ages |
| Zerihun et al., 2011 | Dore Bafeno Health Center, | SNNPR | 1708 | Health facility | Cross-sectional | January 2010. | 269 | febrile outpatients who sought medical attention | All ages | 53% males, 47% females | Microscopy | 178 | 146 | 28 | 4 | All ages |

*(Continued)*

**Table 1.** (Continued)

| Author ID | Study site/ City/district | Region | Altitude (m) | Setting | Study design | Study year/ period | Sample tested | Study population | | | Diagnostic method | Malaria positive | *P. falciparum* | *P. vivax* | Mixed infection | Group |
| | | | | | | | | Key characteristics | Age | Gender | | | | | | |
| Zhou et al., 2016 | Jimma town | Oromia | 1710–1800 | Health facility | Cross-sectional | July 2014 to June 2015 | 1434 | Malaria suspected cases | ND | 48% males, 52% females | Microscopy | 428 | 327 | 97 | 4 | All ages |

*Note: ND* = No data available; SNNPR = Southern Nation and Nationalities People Region; RDT = Rapid Diagnostic Test; PCR = Polymerase Chain Reaction; M = Male, F = Female, Mixed infection: *P.falciparum* and *P. vivax* infection

[1] RDT was also performed in a subset of individuals. Discrepant results between microscopy and RDT were solved by a second microcopy reading

[2] Crude results, not results weighted for HH size

[3] RDT was also performed, but species information is only based on microscopy

[4] Except 2 tests in which RDTs were used

[5] Mixed infections: *P. falciparum* and *P.vivax* (n = 24), and *P.falciparum* and *P. malariae* (n = 4)

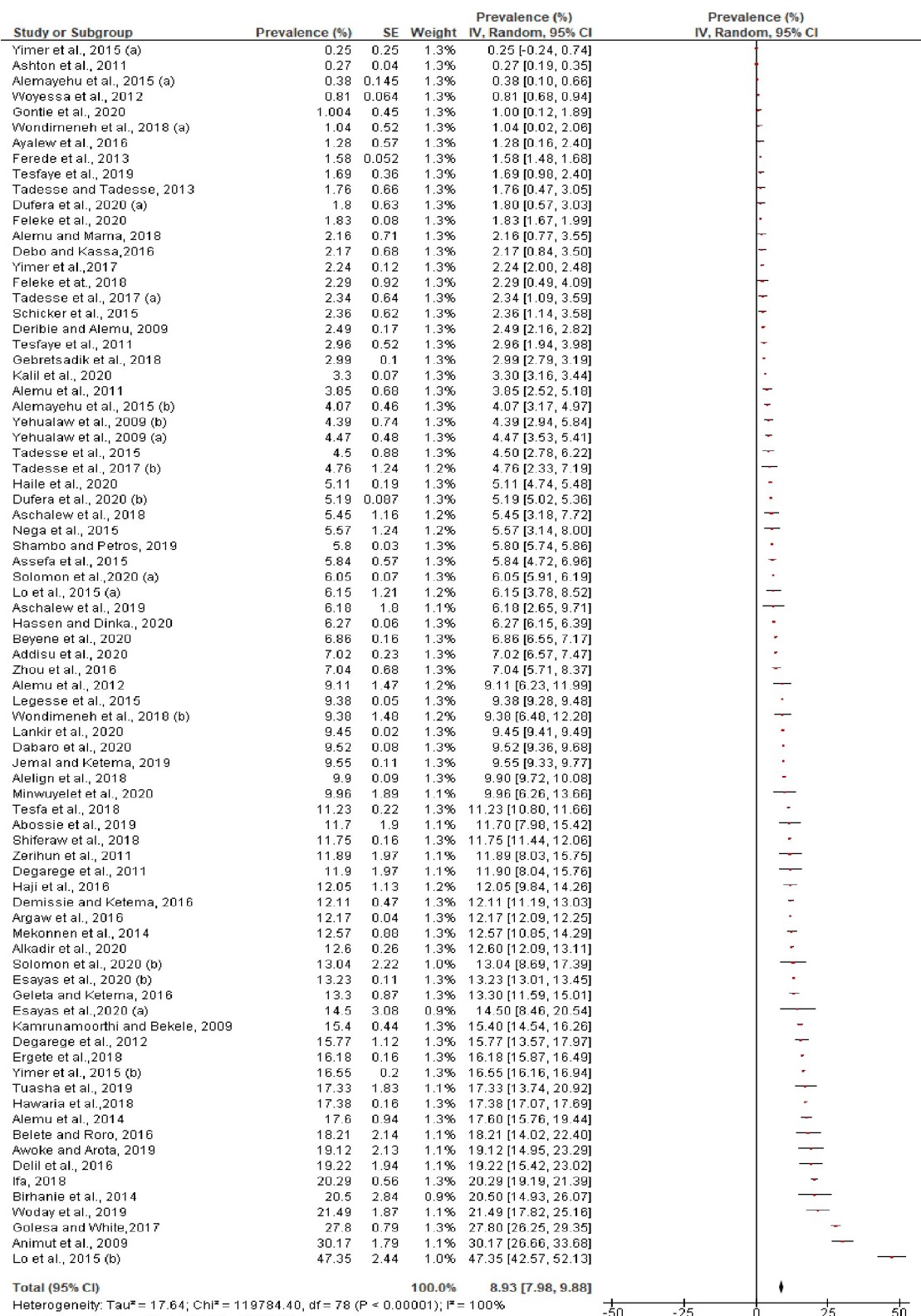

| Study or Subgroup | Prevalence (%) | SE | Weight | Prevalence (%) IV, Random, 95% CI |
|---|---|---|---|---|
| Yimer et al., 2015 (a) | 0.25 | 0.25 | 1.3% | 0.25 [-0.24, 0.74] |
| Ashton et al., 2011 | 0.27 | 0.04 | 1.3% | 0.27 [0.19, 0.35] |
| Alemayehu et al., 2015 (a) | 0.38 | 0.145 | 1.3% | 0.38 [0.10, 0.66] |
| Woyessa et al., 2012 | 0.81 | 0.064 | 1.3% | 0.81 [0.68, 0.94] |
| Gontie et al., 2020 | 1.004 | 0.45 | 1.3% | 1.00 [0.12, 1.89] |
| Wondimeneh et al., 2018 (a) | 1.04 | 0.52 | 1.3% | 1.04 [0.02, 2.06] |
| Ayalew et al., 2016 | 1.28 | 0.57 | 1.3% | 1.28 [0.16, 2.40] |
| Ferede et al., 2013 | 1.58 | 0.052 | 1.3% | 1.58 [1.48, 1.68] |
| Tesfaye et al., 2019 | 1.69 | 0.36 | 1.3% | 1.69 [0.98, 2.40] |
| Tadesse and Tadesse, 2013 | 1.76 | 0.66 | 1.3% | 1.76 [0.47, 3.05] |
| Dufera et al., 2020 (a) | 1.8 | 0.63 | 1.3% | 1.80 [0.57, 3.03] |
| Feleke et al., 2020 | 1.83 | 0.08 | 1.3% | 1.83 [1.67, 1.99] |
| Alemu and Mama, 2018 | 2.16 | 0.71 | 1.3% | 2.16 [0.77, 3.55] |
| Debo and Kassa,2016 | 2.17 | 0.68 | 1.3% | 2.17 [0.84, 3.50] |
| Yimer et al.,2017 | 2.24 | 0.12 | 1.3% | 2.24 [2.00, 2.48] |
| Feleke et al., 2018 | 2.29 | 0.92 | 1.3% | 2.29 [0.49, 4.09] |
| Tadesse et al., 2017 (a) | 2.34 | 0.64 | 1.3% | 2.34 [1.09, 3.59] |
| Schicker et al., 2015 | 2.36 | 0.62 | 1.3% | 2.36 [1.14, 3.58] |
| Deribie and Alemu, 2009 | 2.49 | 0.17 | 1.3% | 2.49 [2.16, 2.82] |
| Tesfaye et al., 2011 | 2.96 | 0.52 | 1.3% | 2.96 [1.94, 3.98] |
| Gebretsadik et al., 2018 | 2.99 | 0.1 | 1.3% | 2.99 [2.79, 3.19] |
| Kalil et al., 2020 | 3.3 | 0.07 | 1.3% | 3.30 [3.16, 3.44] |
| Alemu et al., 2011 | 3.85 | 0.68 | 1.3% | 3.85 [2.52, 5.18] |
| Alemayehu et al., 2015 (b) | 4.07 | 0.46 | 1.3% | 4.07 [3.17, 4.97] |
| Yehualaw et al., 2009 (b) | 4.39 | 0.74 | 1.3% | 4.39 [2.94, 5.84] |
| Yehualaw et al., 2009 (a) | 4.47 | 0.48 | 1.3% | 4.47 [3.53, 5.41] |
| Tadesse et al., 2015 | 4.5 | 0.88 | 1.3% | 4.50 [2.78, 6.22] |
| Tadesse et al., 2017 (b) | 4.76 | 1.24 | 1.2% | 4.76 [2.33, 7.19] |
| Haile et al., 2020 | 5.11 | 0.19 | 1.3% | 5.11 [4.74, 5.48] |
| Dufera et al., 2020 (b) | 5.19 | 0.087 | 1.3% | 5.19 [5.02, 5.36] |
| Aschalew et al., 2018 | 5.45 | 1.16 | 1.2% | 5.45 [3.18, 7.72] |
| Nega et al., 2015 | 5.57 | 1.24 | 1.2% | 5.57 [3.14, 8.00] |
| Shambo and Petros, 2019 | 5.8 | 0.03 | 1.3% | 5.80 [5.74, 5.86] |
| Assefa et al., 2015 | 5.84 | 0.57 | 1.3% | 5.84 [4.72, 6.96] |
| Solomon et al.,2020 (a) | 6.05 | 0.07 | 1.3% | 6.05 [5.91, 6.19] |
| Lo et al., 2015 (a) | 6.15 | 1.21 | 1.2% | 6.15 [3.78, 8.52] |
| Aschalew et al., 2019 | 6.18 | 1.8 | 1.1% | 6.18 [2.65, 9.71] |
| Hassen and Dinka., 2020 | 6.27 | 0.06 | 1.3% | 6.27 [6.15, 6.39] |
| Beyene et al., 2020 | 6.86 | 0.16 | 1.3% | 6.86 [6.55, 7.17] |
| Addisu et al., 2020 | 7.02 | 0.23 | 1.3% | 7.02 [6.57, 7.47] |
| Zhou et al., 2016 | 7.04 | 0.68 | 1.3% | 7.04 [5.71, 8.37] |
| Alemu et al., 2012 | 9.11 | 1.47 | 1.2% | 9.11 [6.23, 11.99] |
| Legesse et al., 2015 | 9.38 | 0.05 | 1.3% | 9.38 [9.28, 9.48] |
| Wondimeneh et al., 2018 (b) | 9.38 | 1.48 | 1.2% | 9.38 [6.48, 12.28] |
| Lankir et al., 2020 | 9.45 | 0.02 | 1.3% | 9.45 [9.41, 9.49] |
| Dabaro et al., 2020 | 9.52 | 0.08 | 1.3% | 9.52 [9.36, 9.68] |
| Jemal and Ketema, 2019 | 9.55 | 0.11 | 1.3% | 9.55 [9.33, 9.77] |
| Alelign et al., 2018 | 9.9 | 0.09 | 1.3% | 9.90 [9.72, 10.08] |
| Minwuyelet et al., 2020 | 9.96 | 1.89 | 1.1% | 9.96 [6.26, 13.66] |
| Tesfa et al., 2018 | 11.23 | 0.22 | 1.3% | 11.23 [10.80, 11.66] |
| Abossie et al., 2019 | 11.7 | 1.9 | 1.1% | 11.70 [7.98, 15.42] |
| Shiferaw et al., 2018 | 11.75 | 0.16 | 1.3% | 11.75 [11.44, 12.06] |
| Zerihun et al., 2011 | 11.89 | 1.97 | 1.1% | 11.89 [8.03, 15.75] |
| Degarege et al., 2011 | 11.9 | 1.97 | 1.1% | 11.90 [8.04, 15.76] |
| Haji et al., 2016 | 12.05 | 1.13 | 1.2% | 12.05 [9.84, 14.26] |
| Demissie and Ketema, 2016 | 12.11 | 0.47 | 1.3% | 12.11 [11.19, 13.03] |
| Argaw et al., 2016 | 12.17 | 0.04 | 1.3% | 12.17 [12.09, 12.25] |
| Mekonnen et al., 2014 | 12.57 | 0.88 | 1.3% | 12.57 [10.85, 14.29] |
| Alkadir et al., 2020 | 12.6 | 0.26 | 1.3% | 12.60 [12.09, 13.11] |
| Solomon et al., 2020 (b) | 13.04 | 2.22 | 1.0% | 13.04 [8.69, 17.39] |
| Esayas et al., 2020 (b) | 13.23 | 0.11 | 1.3% | 13.23 [13.01, 13.45] |
| Geleta and Ketema, 2016 | 13.3 | 0.87 | 1.3% | 13.30 [11.59, 15.01] |
| Esayas et al.,2020 (a) | 14.5 | 3.08 | 0.9% | 14.50 [8.46, 20.54] |
| Kamrunamoorthi and Bekele, 2009 | 15.4 | 0.44 | 1.3% | 15.40 [14.54, 16.26] |
| Degarege et al., 2012 | 15.77 | 1.12 | 1.3% | 15.77 [13.57, 17.97] |
| Ergete et al.,2018 | 16.18 | 0.16 | 1.3% | 16.18 [15.87, 16.49] |
| Yimer et al., 2015 (b) | 16.55 | 0.2 | 1.3% | 16.55 [16.16, 16.94] |
| Tuasha et al., 2019 | 17.33 | 1.83 | 1.1% | 17.33 [13.74, 20.92] |
| Hawaria et al.,2018 | 17.38 | 0.16 | 1.3% | 17.38 [17.07, 17.69] |
| Alemu et al., 2014 | 17.6 | 0.94 | 1.3% | 17.60 [15.76, 19.44] |
| Belete and Roro, 2016 | 18.21 | 2.14 | 1.1% | 18.21 [14.02, 22.40] |
| Awoke and Arota, 2019 | 19.12 | 2.13 | 1.1% | 19.12 [14.95, 23.29] |
| Delil et al., 2016 | 19.22 | 1.94 | 1.1% | 19.22 [15.42, 23.02] |
| Ifa, 2018 | 20.29 | 0.56 | 1.3% | 20.29 [19.19, 21.39] |
| Birhanie et al., 2014 | 20.5 | 2.84 | 0.9% | 20.50 [14.93, 26.07] |
| Woday et al., 2019 | 21.49 | 1.87 | 1.1% | 21.49 [17.82, 25.16] |
| Golesa and White,2017 | 27.8 | 0.79 | 1.3% | 27.80 [26.25, 29.35] |
| Animut et al., 2009 | 30.17 | 1.79 | 1.1% | 30.17 [26.66, 33.68] |
| Lo et al., 2015 (b) | 47.35 | 2.44 | 1.0% | 47.35 [42.57, 52.13] |
| **Total (95% CI)** | | | **100.0%** | **8.93 [7.98, 9.88]** |

Heterogeneity: Tau² = 17.64; Chi² = 119784.40, df = 78 (P < 0.00001); I² = 100%
Test for overall effect: Z = 18.36 (P < 0.00001)

**Fig 3. Individual and pooled estimates of the prevalence of *P. vivax* (mono-infection and mixed infection with *P. falciparum*) in Ethiopia.**

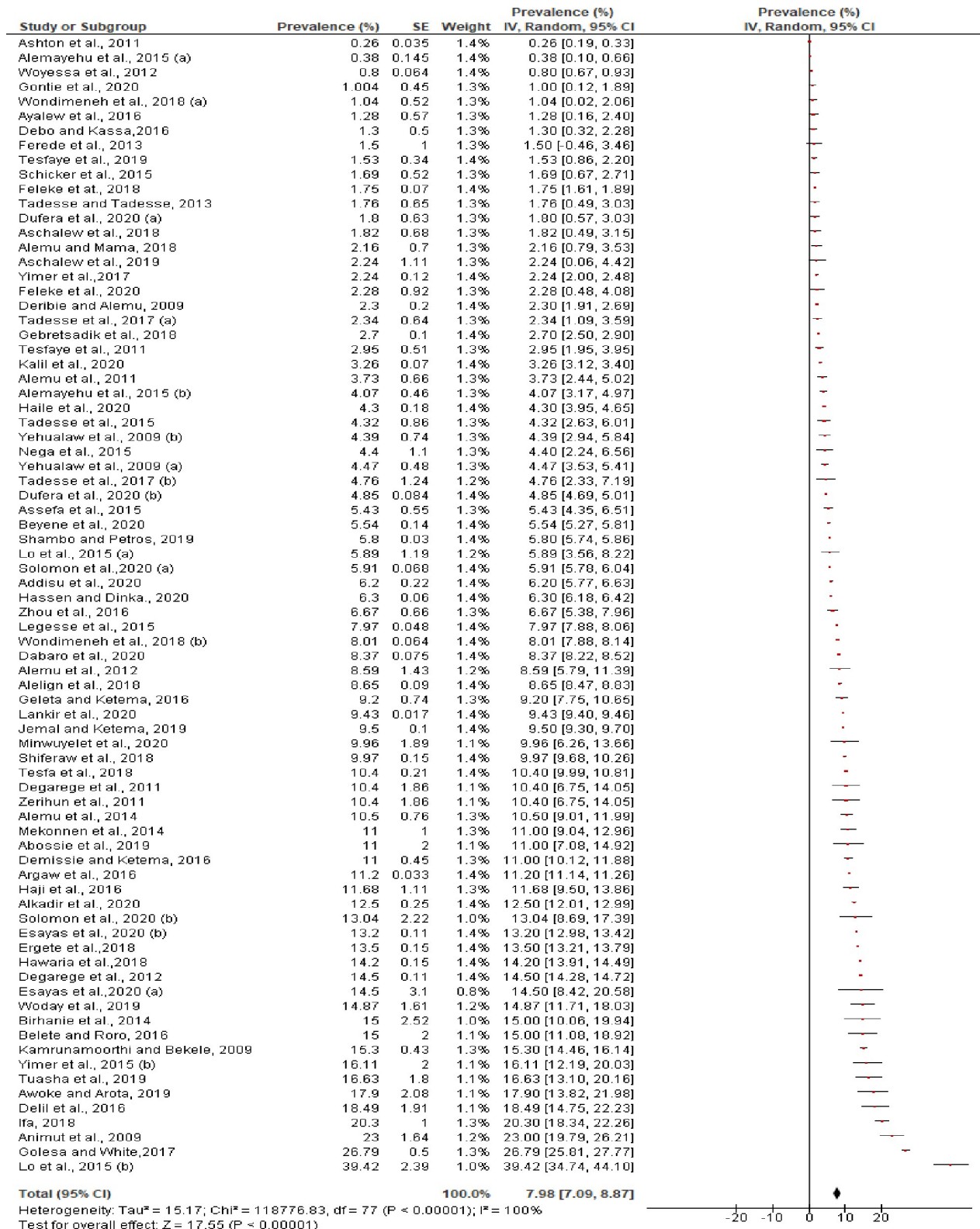

| Study or Subgroup | Prevalence (%) | SE | Weight | Prevalence (%) IV, Random, 95% CI |
|---|---|---|---|---|
| Ashton et al., 2011 | 0.26 | 0.035 | 1.4% | 0.26 [0.19, 0.33] |
| Alemayehu et al., 2015 (a) | 0.38 | 0.145 | 1.4% | 0.38 [0.10, 0.66] |
| Woyessa et al., 2012 | 0.8 | 0.064 | 1.4% | 0.80 [0.67, 0.93] |
| Gontie et al., 2020 | 1.004 | 0.45 | 1.3% | 1.00 [0.12, 1.89] |
| Wondimeneh et al., 2018 (a) | 1.04 | 0.52 | 1.3% | 1.04 [0.02, 2.06] |
| Ayalew et al., 2016 | 1.28 | 0.57 | 1.3% | 1.28 [0.16, 2.40] |
| Debo and Kassa,2016 | 1.3 | 0.5 | 1.3% | 1.30 [0.32, 2.28] |
| Ferede et al., 2013 | 1.5 | 1 | 1.3% | 1.50 [-0.46, 3.46] |
| Tesfaye et al., 2019 | 1.53 | 0.34 | 1.4% | 1.53 [0.86, 2.20] |
| Schicker et al., 2015 | 1.69 | 0.52 | 1.3% | 1.69 [0.67, 2.71] |
| Feleke et al., 2018 | 1.75 | 0.07 | 1.4% | 1.75 [1.61, 1.89] |
| Tadesse and Tadesse, 2013 | 1.76 | 0.65 | 1.3% | 1.76 [0.49, 3.03] |
| Dufera et al., 2020 (a) | 1.8 | 0.63 | 1.3% | 1.80 [0.57, 3.03] |
| Aschalew et al., 2018 | 1.82 | 0.68 | 1.3% | 1.82 [0.49, 3.15] |
| Alemu and Mama, 2018 | 2.16 | 0.7 | 1.3% | 2.16 [0.79, 3.53] |
| Aschalew et al., 2019 | 2.24 | 1.11 | 1.3% | 2.24 [0.06, 4.42] |
| Yimer et al.,2017 | 2.24 | 0.12 | 1.4% | 2.24 [2.00, 2.48] |
| Feleke et al., 2020 | 2.28 | 0.92 | 1.3% | 2.28 [0.48, 4.08] |
| Deribie and Alemu, 2009 | 2.3 | 0.2 | 1.4% | 2.30 [1.91, 2.69] |
| Tadesse et al., 2017 (a) | 2.34 | 0.64 | 1.3% | 2.34 [1.09, 3.59] |
| Gebretsadik et al., 2018 | 2.7 | 0.1 | 1.4% | 2.70 [2.50, 2.90] |
| Tesfaye et al., 2011 | 2.95 | 0.51 | 1.3% | 2.95 [1.95, 3.95] |
| Kalil et al., 2020 | 3.26 | 0.07 | 1.4% | 3.26 [3.12, 3.40] |
| Alemu et al., 2011 | 3.73 | 0.66 | 1.3% | 3.73 [2.44, 5.02] |
| Alemayehu et al., 2015 (b) | 4.07 | 0.46 | 1.3% | 4.07 [3.17, 4.97] |
| Haile et al., 2020 | 4.3 | 0.18 | 1.4% | 4.30 [3.95, 4.65] |
| Tadesse et al., 2015 | 4.32 | 0.86 | 1.3% | 4.32 [2.63, 6.01] |
| Yehualaw et al., 2009 (b) | 4.39 | 0.74 | 1.3% | 4.39 [2.94, 5.84] |
| Nega et al., 2015 | 4.4 | 1.1 | 1.3% | 4.40 [2.24, 6.56] |
| Yehualaw et al., 2009 (a) | 4.47 | 0.48 | 1.3% | 4.47 [3.53, 5.41] |
| Tadesse et al., 2017 (b) | 4.76 | 1.24 | 1.2% | 4.76 [2.33, 7.19] |
| Dufera et al., 2020 (b) | 4.85 | 0.084 | 1.4% | 4.85 [4.69, 5.01] |
| Assefa et al., 2015 | 5.43 | 0.55 | 1.3% | 5.43 [4.35, 6.51] |
| Beyene et al., 2020 | 5.54 | 0.14 | 1.4% | 5.54 [5.27, 5.81] |
| Shambo and Petros, 2019 | 5.8 | 0.03 | 1.4% | 5.80 [5.74, 5.86] |
| Lo et al., 2015 (a) | 5.89 | 1.19 | 1.2% | 5.89 [3.56, 8.22] |
| Solomon et al.,2020 (a) | 5.91 | 0.068 | 1.4% | 5.91 [5.78, 6.04] |
| Addisu et al., 2020 | 6.2 | 0.22 | 1.4% | 6.20 [5.77, 6.63] |
| Hassen and Dinka., 2020 | 6.3 | 0.06 | 1.4% | 6.30 [6.18, 6.42] |
| Zhou et al., 2016 | 6.67 | 0.66 | 1.3% | 6.67 [5.38, 7.96] |
| Legesse et al., 2015 | 7.97 | 0.048 | 1.4% | 7.97 [7.88, 8.06] |
| Wondimeneh et al., 2018 (b) | 8.01 | 0.064 | 1.4% | 8.01 [7.88, 8.14] |
| Dabaro et al., 2020 | 8.37 | 0.075 | 1.4% | 8.37 [8.22, 8.52] |
| Alemu et al., 2012 | 8.59 | 1.43 | 1.2% | 8.59 [5.79, 11.39] |
| Alelign et al., 2018 | 8.65 | 0.09 | 1.4% | 8.65 [8.47, 8.83] |
| Geleta and Ketema, 2016 | 9.2 | 0.74 | 1.3% | 9.20 [7.75, 10.65] |
| Lankir et al., 2020 | 9.43 | 0.017 | 1.4% | 9.43 [9.40, 9.46] |
| Jemal and Ketema, 2019 | 9.5 | 0.1 | 1.4% | 9.50 [9.30, 9.70] |
| Minwuyelet et al., 2020 | 9.96 | 1.89 | 1.1% | 9.96 [6.26, 13.66] |
| Shiferaw et al., 2018 | 9.97 | 0.15 | 1.4% | 9.97 [9.68, 10.26] |
| Tesfa et al., 2018 | 10.4 | 0.21 | 1.4% | 10.40 [9.99, 10.81] |
| Degarege et al., 2011 | 10.4 | 1.86 | 1.1% | 10.40 [6.75, 14.05] |
| Zerihun et al., 2011 | 10.4 | 1.86 | 1.1% | 10.40 [6.75, 14.05] |
| Alemu et al., 2014 | 10.5 | 0.76 | 1.3% | 10.50 [9.01, 11.99] |
| Mekonnen et al., 2014 | 11 | 1 | 1.3% | 11.00 [9.04, 12.96] |
| Abossie et al., 2019 | 11 | 2 | 1.1% | 11.00 [7.08, 14.92] |
| Demissie and Ketema, 2016 | 11 | 0.45 | 1.3% | 11.00 [10.12, 11.88] |
| Argaw et al., 2016 | 11.2 | 0.033 | 1.4% | 11.20 [11.14, 11.26] |
| Haji et al., 2016 | 11.68 | 1.11 | 1.3% | 11.68 [9.50, 13.86] |
| Alkadir et al., 2020 | 12.5 | 0.25 | 1.4% | 12.50 [12.01, 12.99] |
| Solomon et al., 2020 (b) | 13.04 | 2.22 | 1.0% | 13.04 [8.69, 17.39] |
| Esayas et al., 2020 (b) | 13.2 | 0.11 | 1.4% | 13.20 [12.98, 13.42] |
| Ergete et al.,2018 | 13.5 | 0.15 | 1.4% | 13.50 [13.21, 13.79] |
| Hawaria et al.,2018 | 14.2 | 0.15 | 1.4% | 14.20 [13.91, 14.49] |
| Degarege et al., 2012 | 14.5 | 0.11 | 1.4% | 14.50 [14.28, 14.72] |
| Esayas et al.,2020 (a) | 14.5 | 3.1 | 0.8% | 14.50 [8.42, 20.58] |
| Woday et al., 2019 | 14.87 | 1.61 | 1.2% | 14.87 [11.71, 18.03] |
| Birhanie et al., 2014 | 15 | 2.52 | 1.0% | 15.00 [10.06, 19.94] |
| Belete and Roro, 2016 | 15 | 2 | 1.1% | 15.00 [11.08, 18.92] |
| Kamrunamoorthi and Bekele, 2009 | 15.3 | 0.43 | 1.3% | 15.30 [14.46, 16.14] |
| Yimer et al., 2015 (b) | 16.11 | 2 | 1.1% | 16.11 [12.19, 20.03] |
| Tuasha et al., 2019 | 16.63 | 1.8 | 1.1% | 16.63 [13.10, 20.16] |
| Awoke and Arota, 2019 | 17.9 | 2.08 | 1.1% | 17.90 [13.82, 21.98] |
| Delil et al., 2016 | 18.49 | 1.91 | 1.1% | 18.49 [14.75, 22.23] |
| Ifa, 2018 | 20.3 | 1 | 1.3% | 20.30 [18.34, 22.26] |
| Animut et al., 2009 | 23 | 1.64 | 1.2% | 23.00 [19.79, 26.21] |
| Golesa and White,2017 | 26.79 | 0.5 | 1.3% | 26.79 [25.81, 27.77] |
| Lo et al., 2015 (b) | 39.42 | 2.39 | 1.0% | 39.42 [34.74, 44.10] |
| **Total (95% CI)** | | | **100.0%** | **7.98 [7.09, 8.87]** |

Heterogeneity: Tau² = 15.17; Chi² = 118776.83, df = 77 (P < 0.00001); I² = 100%
Test for overall effect: Z = 17.55 (P < 0.00001)

**Fig 4. Individual and pooled estimates of the prevalence of *P. vivax* mono-infection in Ethiopia, 2000–2020.**

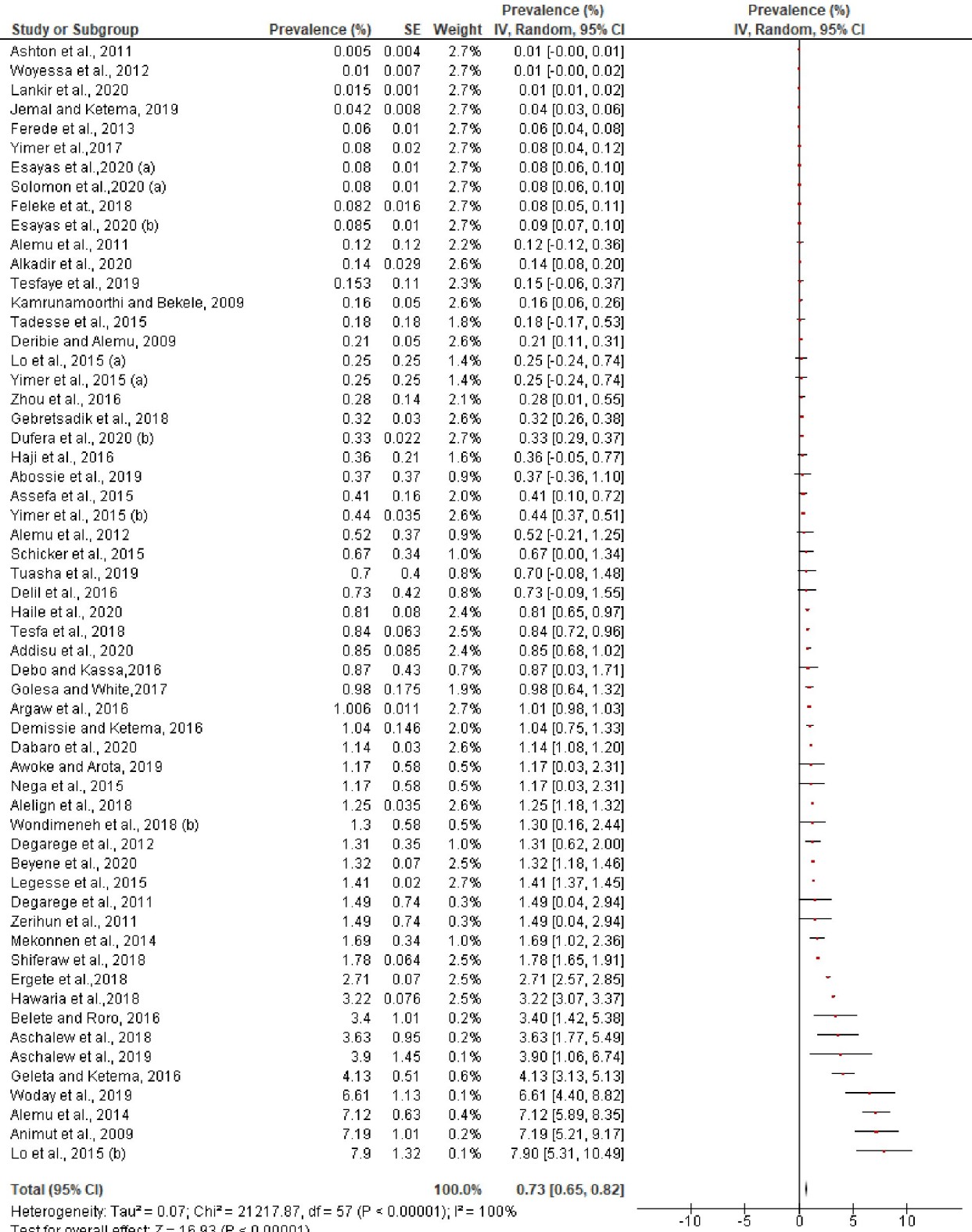

| Study or Subgroup | Prevalence (%) | SE | Weight | Prevalence (%) IV, Random, 95% CI |
|---|---|---|---|---|
| Ashton et al., 2011 | 0.005 | 0.004 | 2.7% | 0.01 [-0.00, 0.01] |
| Woyessa et al., 2012 | 0.01 | 0.007 | 2.7% | 0.01 [-0.00, 0.02] |
| Lankir et al., 2020 | 0.015 | 0.001 | 2.7% | 0.01 [0.01, 0.02] |
| Jemal and Ketema, 2019 | 0.042 | 0.008 | 2.7% | 0.04 [0.03, 0.06] |
| Ferede et al., 2013 | 0.06 | 0.01 | 2.7% | 0.06 [0.04, 0.08] |
| Yimer et al.,2017 | 0.08 | 0.02 | 2.7% | 0.08 [0.04, 0.12] |
| Esayas et al.,2020 (a) | 0.08 | 0.01 | 2.7% | 0.08 [0.06, 0.10] |
| Solomon et al.,2020 (a) | 0.08 | 0.01 | 2.7% | 0.08 [0.06, 0.10] |
| Feleke et at., 2018 | 0.082 | 0.016 | 2.7% | 0.08 [0.05, 0.11] |
| Esayas et al., 2020 (b) | 0.085 | 0.01 | 2.7% | 0.09 [0.07, 0.10] |
| Alemu et al., 2011 | 0.12 | 0.12 | 2.2% | 0.12 [-0.12, 0.36] |
| Alkadir et al., 2020 | 0.14 | 0.029 | 2.6% | 0.14 [0.08, 0.20] |
| Tesfaye et al., 2019 | 0.153 | 0.11 | 2.3% | 0.15 [-0.06, 0.37] |
| Kamrunamoorthi and Bekele, 2009 | 0.16 | 0.05 | 2.6% | 0.16 [0.06, 0.26] |
| Tadesse et al., 2015 | 0.18 | 0.18 | 1.8% | 0.18 [-0.17, 0.53] |
| Deribie and Alemu, 2009 | 0.21 | 0.05 | 2.6% | 0.21 [0.11, 0.31] |
| Lo et al., 2015 (a) | 0.25 | 0.25 | 1.4% | 0.25 [-0.24, 0.74] |
| Yimer et al., 2015 (a) | 0.25 | 0.25 | 1.4% | 0.25 [-0.24, 0.74] |
| Zhou et al., 2016 | 0.28 | 0.14 | 2.1% | 0.28 [0.01, 0.55] |
| Gebretsadik et al., 2018 | 0.32 | 0.03 | 2.6% | 0.32 [0.26, 0.38] |
| Dufera et al., 2020 (b) | 0.33 | 0.022 | 2.7% | 0.33 [0.29, 0.37] |
| Haji et al., 2016 | 0.36 | 0.21 | 1.6% | 0.36 [-0.05, 0.77] |
| Abossie et al., 2019 | 0.37 | 0.37 | 0.9% | 0.37 [-0.36, 1.10] |
| Assefa et al., 2015 | 0.41 | 0.16 | 2.0% | 0.41 [0.10, 0.72] |
| Yimer et al., 2015 (b) | 0.44 | 0.035 | 2.6% | 0.44 [0.37, 0.51] |
| Alemu et al., 2012 | 0.52 | 0.37 | 0.9% | 0.52 [-0.21, 1.25] |
| Schicker et al., 2015 | 0.67 | 0.34 | 1.0% | 0.67 [0.00, 1.34] |
| Tuasha et al., 2019 | 0.7 | 0.4 | 0.8% | 0.70 [-0.08, 1.48] |
| Delil et al., 2016 | 0.73 | 0.42 | 0.8% | 0.73 [-0.09, 1.55] |
| Haile et al., 2020 | 0.81 | 0.08 | 2.4% | 0.81 [0.65, 0.97] |
| Tesfa et al., 2018 | 0.84 | 0.063 | 2.5% | 0.84 [0.72, 0.96] |
| Addisu et al., 2020 | 0.85 | 0.085 | 2.4% | 0.85 [0.68, 1.02] |
| Debo and Kassa,2016 | 0.87 | 0.43 | 0.7% | 0.87 [0.03, 1.71] |
| Golesa and White,2017 | 0.98 | 0.175 | 1.9% | 0.98 [0.64, 1.32] |
| Argaw et al., 2016 | 1.006 | 0.011 | 2.7% | 1.01 [0.98, 1.03] |
| Demissie and Ketema, 2016 | 1.04 | 0.146 | 2.0% | 1.04 [0.75, 1.33] |
| Dabaro et al., 2020 | 1.14 | 0.03 | 2.6% | 1.14 [1.08, 1.20] |
| Awoke and Arota, 2019 | 1.17 | 0.58 | 0.5% | 1.17 [0.03, 2.31] |
| Nega et al., 2015 | 1.17 | 0.58 | 0.5% | 1.17 [0.03, 2.31] |
| Alelign et al., 2018 | 1.25 | 0.035 | 2.6% | 1.25 [1.18, 1.32] |
| Wondimeneh et al., 2018 (b) | 1.3 | 0.58 | 0.5% | 1.30 [0.16, 2.44] |
| Degarege et al., 2012 | 1.31 | 0.35 | 1.0% | 1.31 [0.62, 2.00] |
| Beyene et al., 2020 | 1.32 | 0.07 | 2.5% | 1.32 [1.18, 1.46] |
| Legesse et al., 2015 | 1.41 | 0.02 | 2.7% | 1.41 [1.37, 1.45] |
| Degarege et al., 2011 | 1.49 | 0.74 | 0.3% | 1.49 [0.04, 2.94] |
| Zerihun et al., 2011 | 1.49 | 0.74 | 0.3% | 1.49 [0.04, 2.94] |
| Mekonnen et al., 2014 | 1.69 | 0.34 | 1.0% | 1.69 [1.02, 2.36] |
| Shiferaw et al., 2018 | 1.78 | 0.064 | 2.5% | 1.78 [1.65, 1.91] |
| Ergete et al.,2018 | 2.71 | 0.07 | 2.5% | 2.71 [2.57, 2.85] |
| Hawaria et al.,2018 | 3.22 | 0.076 | 2.5% | 3.22 [3.07, 3.37] |
| Belete and Roro, 2016 | 3.4 | 1.01 | 0.2% | 3.40 [1.42, 5.38] |
| Aschalew et al., 2018 | 3.63 | 0.95 | 0.2% | 3.63 [1.77, 5.49] |
| Aschalew et al., 2019 | 3.9 | 1.45 | 0.1% | 3.90 [1.06, 6.74] |
| Geleta and Ketema, 2016 | 4.13 | 0.51 | 0.6% | 4.13 [3.13, 5.13] |
| Woday et al., 2019 | 6.61 | 1.13 | 0.1% | 6.61 [4.40, 8.82] |
| Alemu et al., 2014 | 7.12 | 0.63 | 0.4% | 7.12 [5.89, 8.35] |
| Animut et al., 2009 | 7.19 | 1.01 | 0.2% | 7.19 [5.21, 9.17] |
| Lo et al., 2015 (b) | 7.9 | 1.32 | 0.1% | 7.90 [5.31, 10.49] |
| **Total (95% CI)** | | | **100.0%** | **0.73 [0.65, 0.82]** |

Heterogeneity: Tau² = 0.07; Chi² = 21217.87, df = 57 (P < 0.00001); I² = 100%
Test for overall effect: Z = 16.93 (P < 0.00001)

**Fig 5. Individual and pooled estimates of the prevalence of mixed infection (*P. vivax* and *P. falciparum*) in Ethiopia, 2000–2020.**

individual studies were inconsistent. However, a significant reduction in the prevalence of *P. vivax* (4.67%, 95%CI: 1.41–7.93%, p<0.0001) was observed in studies conducted at the population /community level (schools, and villages) (**S5 Fig**). Analysis of effects of study years on the pooled estimated prevalence of *P. vivax* revealed lack of statistically significant differences (p = 0.93, $I^2$ = 0%) within the subgroups (**S6 Fig**).

## Meta-regression analysis

A meta-regression analysis was used to determine if sub-groups (geographical situation, altitudes of the study sites, years of the study and study settings) had an effect on the pooled prevalence of *P. vivax* in the country. Findings from this meta-regression analysis further confirmed the effect of the subgroups on the overall pooled *P. vivax* prevalence. Geographical situation of the studies (SNNPR region), study settings (study from health facilities compared to those from community), and studies reported from areas whose altitude ranges from 1500-1750m seemed to be associated with a significant increasing in the prevalence of *P. vivax* malaria in Ethiopia, but the remaining variables such as study year did not show significant effect on the pooled prevalence of *P. vivax*. Studies from altitude ranges from 2000 to 2500m showed comparatively higher prevalence of *P, vivax* next to altitude range from 1500–1750, although significant difference was not observed (Table 2).

## Discussion

This study aimed to review the overall prevalence of *P. vivax* malaria infections in Ethiopia. For this purpose, any study that investigated the prevalence and epidemiology of malaria in the country, and which contained detailed data on *P. vivax* was included. The overall pooled prevalence of *P. vivax* malaria (mono-infection or mixed infection among symptomatic and asymptomatic patients) in Ethiopia was 8.93% (95% CI: 7.98–9.88%). Prevalence among *P. vivax* mono-infection alone was 7.98% (95% CI: 7.09–8.87%). These figures are much higher than the predicted endemicity values of *P. vivax* prevalence for Madagascar and Ethiopia, and parts of South Sudan and Somalia, which rarely exceed 2% [87]. Typically, the *P. vivax* parasite load in peripheral blood is very low as compared to *P. falciparum*, often hindering its diagnosis using conventional optic microscopy [88]. However, such low-level parasitemias are sufficient to act as reservoirs and sustain transmission of the parasite [89]. Although microscopy is still the gold standard tool for malaria diagnosis in Ethiopia, a more accurate approach for diagnosis would require the use of more sensitive techniques such as PCR or LAMP, capable of detecting submicroscopic carriage and mixed infections in areas where the two main parasites

**Table 2. Meta-regression analysis of impact of subgroups on prevalence of *P. vivax* in Ethiopia, 2000–2020.**

| Subgroup | Covariate | Coefficient | SE | 95% Lower CI | 95% Upper CI | Z-value | P-value |
|---|---|---|---|---|---|---|---|
| | **Intercept** | **8.05** | **1.45** | **5.21** | **10.895** | **5.55** | **0.00** |
| Region | Oromia | 0.65 | 1.09 | -1.45 | 2.79 | 0.59 | 0.55 |
| | SNNPR | 2.60 | 1.02 | 0.6 | 4.61 | 2.54 | 0.01 |
| Altitude | 1500-1750m | 3.30 | 1.44 | 0.49 | 6.11 | 2.3 | 0.02 |
| | 1750-2000m | -0.31 | 1.35 | -2.95 | 2.33 | -0.23 | 0.82 |
| | 2000-2500m | 2.81 | 1.68 | -0.5 | 6.11 | 1.67 | 0.09 |
| | Mix | 2.56 | 1.11 | 0.38 | 4.74 | 2.3 | 0.02 |
| Study setting | Community | -5.94 | 1.004 | -7.91 | -3.97 | -5.91 | 0.00 |
| Study year | After 2010 | -0.88 | 1.12 | -3.07 | 1.31 | -0.79 | 0.43 |

Note: CI = confidence interval, SE = standard error.

(*P. falciparum* and *P. vivax*) co-exist [90]. Given that most of the studies included in this review used microscopy as the chosen diagnostic tool, it is likely that the reported prevalence rates are an underestimate of the true prevalence of this parasite.

Ethiopia has variable topographic features that govern the distribution of malaria infection. Generally, it is agreed that malaria is endemic in areas with altitude lower than1500m (lowlands with seasonal/intense transmission) and rare in areas above 2000m (highland with occasional epidemic) [91]. However, in contrast to the general assumption, some studies reporting data from the highlands known for occasional malaria epidemics were found to contribute for a higher prevalence (9.80%, 95%CI: 6.73–12.87%) of *P. vivax*. This might be attributed to its survival ability in colder climate than other *Plasmodium* species [92]. A recent nationwide malaria epidemiological and interventional survey report confirms this finding, establishing the expansion of malaria to areas with altitude higher than 2000m [14], which were previously considered malaria free zones [93] and re-classified them as with moderate annual parasite incidence (APIs). The same report further indicated this as a new risk factor interfering with the current national malaria interventional activities [14]. A sero-prevalence study further strengthened the lack of significant differences in the transmission of *P. vivax* due to altitudinal variation (below or above 2000m) [93]. Rather, *P. vivax* showed direct relation with increasing elevation among children aged <5 years and high sero-positivity (20.9, 95% CI: 17.4–24.9) was observed at higher elevations [93]. The increasing evidence on the transmission of *P. vivax* in the areas traditionally considered as malaria free is an indication of the expansion of malaria transmission in Ethiopia to higher altitude settings. This expansion might be attributed to different developmental plans such as dam constructions, and the use of river water for irrigation purposes, deforestation, population pressures, and lack of appropriate environmental management system [86, 94], which could cause local environmental modifications contributing to the creation of new suitable vector breeding sites or expansion of mosquito's habitat to non-endemic regions; besides changing human settlement pattern [95]. Malaria is one of the most climate sensitive diseases [96, 97] with significant associations between malaria incidence and temperature [96], relative humidity [97, 98] and rainfall [99], all of which do play a significant role in malaria transmission, which makes the vector controlling efforts very challenging. In addition, there are several Anopheles species with some different complexes, thus facilitating transmission into different ecological niches [100]. Furthermore, unlike other plasmodium species, *P. vivax* is capable of undergoing sporogonic development in the mosquito at lower temperatures [101] and able to expand to the highland areas. Growing evidence on *P. vivax* malaria distribution across other areas of Sub-Saharan Africa has further revealed that *P. vivax* appears to become proportionally more significant where overall malaria prevalence is lower [9].

Regional variation on *P. vivax* malaria prevalence was observed in the current review. In very recent years, significant reduction in *P. vivax* malaria burden has been predominantly observed in the Oromia region, as compared to the other regions [19, 72]. According to the National Strategic Plan for Malaria Prevention, Control and Elimination in Ethiopia, the malaria burden was significantly reduced over three survey years (2007, 2011 and 2015) with 0.3% nationwide prevalence in the year 2015 [90]. This figure is relatively lower than reports made from other regions including SNNPR (0.5%), Amhara (0.8%), Benshangul (2.7%) and Gambella (6%) in the same year [90].

Compared to the national report, the prevalence of *P. vivax* malaria infection reported in the current review is much higher. This is due to the fact that the national report was the overall national malaria prevalence, which included only recent data (after malarial morbidity and mortality burden started decreasing) from all malaria transmission settings (low, middle, and high). But, this review only focused on prevalence of *P. vivax* malaria infection and included

almost all studies conducted at high malaria transmission areas, and the prevalence data of 20 years. The recent national sero-prevalence analysis by region supports this finding, with lower *P. vivax* sero-prevalence documented in Oromia than in Amhara (36.7% (95% CI: 30.0–44.1) and SNNPR regions [92], although the detected antibodies might not correspond adequately to the existing infection prevalence.

Following the rise in malaria prevalence as observed in the year 2010/11, the deployment of malaria interventions already initiated in Ethiopia was boosted. This included the distribution of free ITN, IRS, and RDTs as a supplement for malaria diagnosis in remote areas, and the scale-up of ACT deployment and training of health extension workers [102]. As a result, the overall national malaria burden decreased from 0.5% prevalence in 2011 to 0.3% in 2015 [90]. Our meta-analysis on studies whose survey years were before and after the scaling-up of national malaria intervention activities did not show significant effect on the pooled estimated prevalence of *P. vivax* in Ethiopia. However, results from meta-regression indicated that prevalence of *P. vivax* observed after the scaling up of the interventional activities in Ethiopia, showed significant reduction. This finding is in agreement with the global *P. vivax* malaria burden reduction observed (41.6% reduction from 2000 to 2017) in most endemic areas [103]. Although the trend showed a declining pattern, burden due to *P. vivax* in Ethiopia appears considerable, and will cause enormous challenges, calling for careful regular surveillance by concerned bodies. Mainly it's apparent complex parasite biology, pathophysiology, treatment response, the raising problem of Duffy negative individuals that are now infected by *P. vivax* and transmission patterns [104] will make its future eradication goal very challenging. In addition, the hypnozoite's dormant liver stages, responsible for the potential repeated relapses that can occur within weeks, months, or many years after the initial inoculation, blur our current understanding of *P. vivax* epidemiology, and will not be affected unless specific radical cure is conducted [102]. In the absence of such anti-hypnozoite drugs, the current first line drugs used in Ethiopia for *P.vivax* malaria, be it chloroquine or other artemisinin based-combination therapies, will not affect the liver stage hypnozoites [9], thus hindering its adequate control. In addition, ITN and IRSs currently in use might not be efficient in completely preventing new infection, in general, and the relapse from liver stages in particular [9], mosquito species that transmit *P. vivax* bite mostly outdoors and which also changed its biting time from midnight to dawn [105]. Some populations of *An. arabiensis* were reported to even avoid fatal insecticide exposure [106, 107].

## Strengths and limitations of the study

To the best of our knowledge, this is the first detailed systematic review and meta-analysis of only *P. vivax* epidemiology in Ethiopia that included facility and community level studies. A recent systematic review and meta-analysis by Deresse and Girma, [108] assessed (using 35 studies) the prevalence of *P. falciparum* and *P. vivax* in Ethiopia and found 25.8% prevalence all together. Its main objective was to show a general picture on malaria prevalence in Ethiopia. Hence *P. vivax* prevalence/epidemiology was not uniquely reviewed, analyzed or presented separately in the study. Furthermore, the study didn't include the major databases such as Web of Science, Scopus, and EMBASE, but only retrieved articles from PubMed and Google scholar. In addition, it did not assess the role of subgroups such as location, eco-epidemiological zones, study setting and survey years, on the overall pooled prevalence of malaria, in general, and *P. vivax* in particular. The omission of subgroups appears to have significant impact, given that these subgroups showed a significant role on the estimated prevalence of *P. vivax* in our analysis. Hence, the strength of this review is the fact that it included many other new studies to date (n = 44) on *P. vivax* in Ethiopia besides the 35 studies included in the previous review and portrayed the epidemiological distribution of *P. vivax* nationwide [108]

The major limitation of this review was that about one third of the included studies depended on data extracted from retrospective medical case records, reviewed to investigate the prevalence and trends of malaria. Although case record reviews are the most universally used method for prevalence studies, it is often challenging to obtain, in a standardized way, all required data about the individual patient, including socio-demographic and clinical data, how target groups were identified, recruited and the exact diagnostic tools used at the time of enrollment of each participant. In addition, for some of the studies included in the review, their main objective was not set to assess the prevalence or geographical distribution or epidemiological trends of malaria. Some were designed to show association between malaria prevalence and ABO blood groups/helminthic infection/HIV infection/ ITN utilization /hematological profile of malaria patients/ drug efficacy evaluation against *P. vivax*/or comparative evaluation of different malaria diagnostic tests or tools (microscopy Vs PCR). Data from this kind of studies often don't allow an adequate evaluation of the quality criteria set for prevalence/observational studies. Thus, they were included in the review only if they contained data on prevalence of malaria and different *Plasmodium* species. Moreover, significant heterogeneity of the eligible studies observed in this review may require further analysis. Finally, the exclusion of unpublished studies as well as interventional studies may lead, potentially, to loss of substantial data.

## Conclusions

The overall estimated prevalence of *P. vivax* was 8.93% (95%CI: 7.98–9.88). Most of the studies included in the current review met the quality criteria and there was no publication bias. This parasite has historically been widely distributed in the central west region of Ethiopia, and is now steadily extending to the North West and South West regions of the country. Oromia, Amhara and SNNPR are the three major regions where *P. vivax* has spread predominantly with wide-ranging prevalence. *P. vivax* epidemiology has shown the trend of expansion to the highland, causing occasional malaria epidemics, although the existing deployed interventions seem to have an impact on prevalence of this parasite.

## Supporting information

**S1 Table. Summary of search keywords/terms.**
(DOCX)

**S2 Table. Excluded studies and reasons for exclusion of studies on prevalence of *P. vivax* infection in Ethiopia.**
(DOCX)

**S3 Table. Risk bias assessment based on the Prevalence Critical Appraisal Instrument of studies on prevalence of *P. vivax* infection in Ethiopia.**
(DOCX)

**S1 Fig. Boxplot of studies on prevalence of *P. vivax* infection in Ethiopia.**
(DOCX)

**S2 Fig. Funnel plot for publication bias assessment of studies on prevalence of *P. vivax* infection in Ethiopia.**
(DOCX)

**S3 Fig. Pooled estimates of prevalence of *P. vivax* for different locations/regions of Ethiopia.**
(DOCX)

**S4 Fig. Estimate prevalence of P. vivax in different eco-epidemiological zones of Ethiopia.**
(DOCX)

**S5 Fig. Prevalence of *P. vivax* at different study settings in Ethiopia.**
(DOCX)

**S6 Fig. Prevalence of *P. vivax* with respect to year of survey in Ethiopia.**
(DOCX)

## Acknowledgments

Authors of the study would like to thank the staff members at Jimma University main library and ISGlobal, Institute for Global Health, Hospital Clinic-Universitat de Barcelona, Barcelona, Spain for the enormous support obtained during study identification and screening.

## Author Contributions

**Conceptualization:** Tsige Ketema, Ketema Bacha, Hernando A. del Portillo, Quique Bassat.

**Data curation:** Tsige Ketema, Kefelegn Getahun, Quique Bassat.

**Formal analysis:** Tsige Ketema.

**Methodology:** Tsige Ketema, Ketema Bacha, Kefelegn Getahun, Hernando A. del Portillo, Quique Bassat.

**Project administration:** Tsige Ketema, Quique Bassat.

**Software:** Tsige Ketema, Ketema Bacha, Kefelegn Getahun.

**Supervision:** Tsige Ketema, Ketema Bacha, Hernando A. del Portillo, Quique Bassat.

**Validation:** Ketema Bacha, Quique Bassat.

**Writing – original draft:** Tsige Ketema, Ketema Bacha.

**Writing – review & editing:** Tsige Ketema, Ketema Bacha, Kefelegn Getahun, Hernando A. del Portillo, Quique Bassat.

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
