## [Decision Letter · Decision Letter 0]

15 Jul 2021

Dear Dr Ketema,

Thank you very much for submitting your manuscript "Plasmodium vivax epidemiology in Ethiopia 2000-2020: a systematic review and meta-analysis" for consideration at PLOS Neglected Tropical Diseases. As with all papers reviewed by the journal, your manuscript was reviewed by members of the editorial board and by several independent reviewers. The reviewers appreciated the attention to an important topic. Based on the reviews, we are likely to accept this manuscript for publication, providing that you modify the manuscript according to the review recommendations. 

Sincerely,

Paul O. Mireji, PhD

Associate Editor

Hans-Peter Fuehrer

Deputy Editor

Reviewer's Responses to Questions

**Key Review Criteria Required for Acceptance?**

**Methods**

-Are the objectives of the study clearly articulated with a clear testable hypothesis stated?

-Is the study design appropriate to address the stated objectives?

-Is the population clearly described and appropriate for the hypothesis being tested?

-Is the sample size sufficient to ensure adequate power to address the hypothesis being tested?

-Were correct statistical analysis used to support conclusions?

-Are there concerns about ethical or regulatory requirements being met?

Reviewer #1: The objectives of the study are clearly stated. The study does not require a hypothesis.

Reviewer #2: The study design and analytical approach were appropriate

**Results**

-Does the analysis presented match the analysis plan?

-Are the results clearly and completely presented?

-Are the figures (Tables, Images) of sufficient quality for clarity?

Reviewer #1: The analysis fits the analysis plan, the results are clearly presented in appropriate tables.

Reviewer #2: the results are clearly presented

**Conclusions**

-Are the conclusions supported by the data presented?

-Are the limitations of analysis clearly described?

-Do the authors discuss how these data can be helpful to advance our understanding of the topic under study?

-Is public health relevance addressed?

Reviewer #1: The results support the conclusions and the limitations of the study and analysis are well described. The authors have indicated how their results have expanded knowledge of P vivax in Ethiopia. They have also indicated the relevance of their results to public health.

Reviewer #2: Yes the conclusions are supported by the results and discussions are important in the field of vivax malaria epidemiology. 

The only error observed was for some reference e.g. ref 95 Kibret et al., 2014 that does not discuss vivax per se but rather the malaria vector. It is important that all references are correctly cited

**Editorial and Data Presentation Modifications?**

Reviewer #1: The paper would benefit by including a sketch map showing regions of high and low prevalence and also areas where P vivax is expanding in Ethiopia.

Reviewer #2: minor revision

**Summary and General Comments**

Reviewer #1: Plasmodium vivax is a serious piblic health problem in Ethiopis and neighbouring countries. The prevalence of the disease is subject to change due to drug resistance, or ineffective vector control. Furthermore, vectotor populations may change due to environmental change and climate change, it is therefore important to take stock of the disease prevalence from time to time.

This can be done by analysing historical and contemporary published data in a systematic fashion and in accordance withe the rules of meta-analysis. This study was carried out in accordance withd these rules in order to arrive at unbiased conclusions.

The results of the study provide reliable estimates of the prevalence of P. vivax in Ethiopia. The results have been adequately discussed and the authors have confined their conclusions on the evidence they have gathered.

Neverthelessless they may make reference to the potential effect of changes in vector ecology and in particular to environmental change. Vectors such as Anophrles arabiensis, Anopheles coustani and Anopheles pharoensis are exophilic and partially anthropophagic making their control difficult. The authors may wish to address this issue in the discussion.

Reviewer #2: The manuscript is well written and follows a standard systematic review process for PRISMA. The data presented is of importance in understand vivax malaria in Africa over time and potential implications in the current and future malaria control efforts especially for East African regions and a now in West African region. It is also worth noting vivax malaria has potential to spread further in Africa due to parasite adaptation to explore alternate invasion pathways other than Duffy binding ligand. Therefore, this makes it of paramount importance to malaria control programmes in Africa

PLOS authors have the option to publish the peer review history of their article (what does this mean?). If published, this will include your full peer review and any attached files.

Reviewer #1: Yes: Dr. Andrew K. Githeko PhD

Reviewer #2: No

Figure Files:

Data Requirements:

Reproducibility:

References

---

## [Decision Letter · Decision Letter 1]

1 Sep 2021

Dear Dr Ketema,

We are pleased to inform you that your manuscript 'Plasmodium vivax epidemiology in Ethiopia 2000-2020: a systematic review and meta-analysis' has been provisionally accepted for publication in PLOS Neglected Tropical Diseases.

Best regards,

Paul O. Mireji, PhD

Associate Editor

Hans-Peter Fuehrer

Deputy Editor

Reviewer's Responses to Questions

**Key Review Criteria Required for Acceptance?**

**Methods**

-Are the objectives of the study clearly articulated with a clear testable hypothesis stated?

-Is the study design appropriate to address the stated objectives?

-Is the population clearly described and appropriate for the hypothesis being tested?

-Is the sample size sufficient to ensure adequate power to address the hypothesis being tested?

-Were correct statistical analysis used to support conclusions?

-Are there concerns about ethical or regulatory requirements being met?

Reviewer #1: As in prevjouse review.

Reviewer #2: as previously noted the manuscript followed standard procedures with clear hypothesis, objectives and methods

**Results**

-Does the analysis presented match the analysis plan?

-Are the results clearly and completely presented?

-Are the figures (Tables, Images) of sufficient quality for clarity?

Reviewer #1: As in ptdviouse review.

Reviewer #2: The analysis were correctly done and result clearly and completely presented

**Conclusions**

-Are the conclusions supported by the data presented?

-Are the limitations of analysis clearly described?

-Do the authors discuss how these data can be helpful to advance our understanding of the topic under study?

-Is public health relevance addressed?

Reviewer #1: As in pfdviouse review. Conluxions are OK.

Reviewer #2: results supports the conclusions

**Editorial and Data Presentation Modifications?**

Reviewer #1: No changes reauired

Reviewer #2: no additional concerns

**Summary and General Comments**

Reviewer #1: No changes required

Reviewer #2: the issues raised before were minor and have now been corrected hence I support the manuscript publication

PLOS authors have the option to publish the peer review history of their article (what does this mean?). If published, this will include your full peer review and any attached files.

Reviewer #1: **Yes: **Dr Andrew K. Githeko PhD

Reviewer #2: No

---

## [Editor Report · Acceptance letter]

8 Sep 2021

Dear Dr Ketema,

We are delighted to inform you that your manuscript, "Plasmodium vivax epidemiology in Ethiopia 2000-2020: a systematic review and meta-analysis," has been formally accepted for publication in PLOS Neglected Tropical Diseases.

Best regards,

Shaden Kamhawi

co-Editor-in-Chief

Paul Brindley

co-Editor-in-Chief
